# DREAMFLOW: HIGH-QUALITY TEXT-TO-3D GENERATION BY APPROXIMATING PROBABILITY FLOW

**Kyungmin Lee**[1]  **Kihyuk Sohn**[2]  **Jinwoo Shin**[1]
[1]KAIST  [2]Google Research
[1]{kyungmnlee, jinwoos}@kaist.ac.kr  [2]kihyuks@google.com

## ABSTRACT

Recent progress in text-to-3D generation has been achieved through the utilization of score distillation methods: they make use of the pre-trained text-to-image (T2I) diffusion models by distilling via the diffusion model training objective. However, such an approach inevitably results in the use of random timesteps at each update, which increases the variance of the gradient and ultimately prolongs the optimization process. In this paper, we propose to enhance the text-to-3D optimization by leveraging the T2I diffusion prior in the generative sampling process with a predetermined timestep schedule. To this end, we interpret text-to-3D optimization as a multi-view image-to-image translation problem, and propose a solution by approximating the probability flow. By leveraging the proposed novel optimization algorithm, we design DreamFlow, a practical three-stage coarse-to-fine text-to-3D optimization framework that enables fast generation of high-quality and high-resolution (i.e., $1024 \times 1024$) 3D contents. For example, we demonstrate that DreamFlow is 5 times faster than the existing state-of-the-art text-to-3D method, while producing more photorealistic 3D contents.[1]

## 1 INTRODUCTION

High-quality 3D content generation is crucial for a broad range of applications, including entertainment, gaming, augmented/virtual/mixed reality, and robotics simulation. However, the current 3D generation process entails tedious work with 3D modeling software, which demands a lot of time and expertise. Thereby, 3D generative models (Gao et al., 2022; Chan et al., 2022; Zeng et al., 2022) have brought large attention, yet they are limited by their generalization capability to creative and artistic 3D contents due to the scarcity of high-quality 3D dataset.

Recent works have demonstrated the great promise of text-to-3D generation, which enables creative and diverse 3D content creation with textual descriptions (Jain et al., 2022; Mohammad Khalid et al., 2022; Poole et al., 2022; Lin et al., 2023; Wang et al., 2023b; Tsalicoglou et al., 2023). Remarkably, those approaches do not exploit any 3D data, while relying on the rich generative prior of text-to-image diffusion models (Nichol et al., 2021; Saharia et al., 2022; Balaji et al., 2022; Rombach et al., 2022). On this line, DreamFusion (Poole et al., 2022) first proposed to optimize a 3D representation such that the image rendered from any view is likely to be that of sampled from reside in the high-density region of the pre-trained diffusion model. To this end, they introduced Score Distillation Sampling, which distills the pre-trained knowledge by the diffusion training objective (Hyvärinen & Dayan, 2005). Since DreamFusion demonstrates the great potential of text-to-3D generation, subsequent studies have improved this technology by using different 3D representations and advanced score distillation methods (Lin et al., 2023; Wang et al., 2023b).

However, score distillation methods exhibit a high-variance gradient, which requires a lengthy optimization process to optimize a 3D representation. This limits the scalability to the usage of larger diffusion priors to generate high-quality and high-resolution 3D content (Podell et al., 2023). This is in part due to the formulation of the score distillation method that aims to distill the diffusion prior of all noise level (i.e., Eq. 7), where the noise timesteps are randomly drawn at each update. In contrast, conventional 2D diffusion generative processes implement a predetermined noise schedule

---

[1]Visit project page for visualizations of our method.

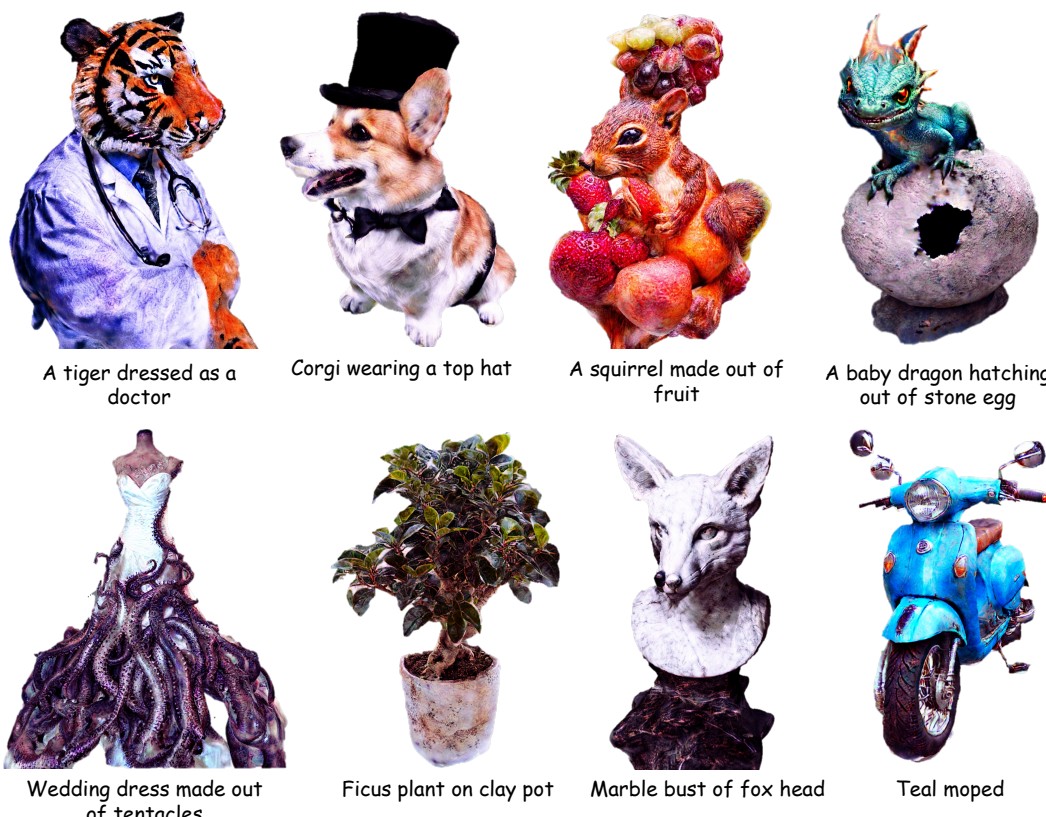

A tiger dressed as a doctor     Corgi wearing a top hat     A squirrel made out of fruit     A baby dragon hatching out of stone egg

Wedding dress made out of tentacles     Ficus plant on clay pot     Marble bust of fox head     Teal moped

Figure 1: **Examples of 3D scene generated by DreamFlow**. DreamFlow can generate photorealistic 3D models from text prompts with reasonable generation time (e.g., less than 2 hours), which have been possible by elucidated optimization strategy using generative diffusion priors.

that gradually transports the noise to the data distribution. This leads us to the following natural question: How to emulate the diffusion generative process for text-to-3D generation?

In this paper, we propose an efficient text-to-3D optimization scheme that aligns with the diffusion generative process. To be specific, unlike the score distillation sampling that uses the training loss of the diffusion model as an objective, our method makes use of the diffusion prior in the generative (or sampling) process, by approximating the reverse generative probability flow (Song et al., 2020b;a). In particular, we have framed the text-to-3D optimization as a multi-view image-to-image translation problem, where we use Schrödinger Bridge problem (Schrödinger, 1932) to derive its solution with probability flow. Then, we propose to match the trajectory of the probability flow to that from the pre-trained text-to-image diffusion model to effectively utilize its rich knowledge. Lastly, we approximate the probability flow to cater to text-to-3D optimization and conduct amortized sampling to optimize multi-view images of a 3D representation. Here, our approach implements a predetermined noise schedule during text-to-3D optimization, as the generative 2D diffusion model does. With our new optimization algorithm, we additionally present a practical text-to-3D generation framework, dubbed **DreamFlow**, which generates high-quality and high-resolution 3D content (see Figure 1). Our approach is conducted in a coarse-to-fine manner, where we generate NeRF, extract and fine-tune the 3D mesh, and refine the mesh with high-resolution diffusion prior (see Figure 3).

Through experiments on human preference studies, we demonstrate that DreamFlow provides the most photorealistic 3D content compared to existing methods including DreamFusion (Poole et al., 2022), Magic3D (Lin et al., 2023), and ProlificDreamer (Wang et al., 2023b). We also show that DreamFlow outperforms ProlificDreamer with respect to CLIP (Radford et al., 2021) R-precision score (in both NeRF generation and 3D mesh fine-tuning), while being 5× faster in generation.

## 2 PRELIMINARY

We briefly introduce the score-based generative models (SGMs) in the general context, which encompasses the diffusion models (DMs). Given the data distribution $p_{\text{data}}(\boldsymbol{x})$, diffusion process starts off by diffusing $\boldsymbol{x}_0 \sim p_{\text{data}}(\boldsymbol{x}_0)$ with forward stochastic differential equation (SDE) and the generative process is followed by reverse SDE for time $t \in [0, T]$, given as follows:

$$\mathrm{d}\boldsymbol{x} = \boldsymbol{f}(\boldsymbol{x}, t)\mathrm{d}t + g(t)\mathrm{d}\boldsymbol{w}, \quad \mathrm{d}\boldsymbol{x} = \left[\boldsymbol{f}(\boldsymbol{x}, t) - g^2(t)\nabla_{\boldsymbol{x}}\log p_t(\boldsymbol{x})\right]\mathrm{d}t + g(t)\mathrm{d}\boldsymbol{w}, \quad (1, 2)$$

where $\boldsymbol{f}(\cdot, \cdot)$ and $g(\cdot)$ are the drift and diffusion coefficient, respectively, $\boldsymbol{w}$ is a standard Wiener process, and $\nabla_{\boldsymbol{x}}\log p_t(\boldsymbol{x})$ is a score function of the marginal density from Eq. 1, where $p_0(\boldsymbol{x}) = p_{\text{data}}(\boldsymbol{x})$. Interestingly, Song et al. (2020b) show that there is a deterministic ordinary differential equation (ODE), referred as *Probability Flow ODE* (PF ODE), which has same marginal density to $p_t(\boldsymbol{x})$ throughout its trajectory. The probability flow ODE is given as follows:

$$\mathrm{d}\boldsymbol{x} = \left[\boldsymbol{f}(\boldsymbol{x}, t) - \frac{1}{2}\sigma^2(t)\nabla_{\boldsymbol{x}}\log p_t(\boldsymbol{x})\right]\mathrm{d}t. \quad (3)$$

Here, the forward SDE is designed that $p_T(\boldsymbol{x})$ is sufficiently close to a tractable Gaussian distribution, so that one can sample from data distribution by reverse SDE or PF ODE. In particular, we follow the setup of (Karras et al., 2022) to formulate the diffusion model, where we let $\boldsymbol{f}(\boldsymbol{x}, t) = \boldsymbol{0}$, $g(t) = \sqrt{2\dot{\sigma}(t)\sigma(t)}$ for a decreasing noise schedule $\sigma : [0, T] \to \mathbb{R}_+$. Also, we denote $p(\boldsymbol{x}; \sigma)$ be the smoothed distribution by adding i.i.d Gaussian noise of standard deviation $\sigma$. Then the evolution of a sample $x_1 \sim p(x_1; \sigma(t_1))$ from time $t_1$ to $t_2$ yields $\boldsymbol{x}_2 \sim p(\boldsymbol{x}_2; \sigma(t_2))$ with following PF ODE:

$$\mathrm{d}\boldsymbol{x} = -\dot{\sigma}(t)\,\sigma(t)\,\nabla_{\boldsymbol{x}}\log p\big(\boldsymbol{x}; \sigma(t)\big)\mathrm{d}t, \quad (4)$$

where dot denotes time-derivative. In practice, we solve ODE by taking finite steps over discrete time schedule, where the derivative is evaluated at each timestep (Karras et al., 2022; Song et al., 2020a).

**Training diffusion models.** The diffusion model trains a neural network that approximates a score function of each $p(x; \sigma)$ by using *Denoising Score Matching* (DSM) (Hyvärinen & Dayan, 2005) objective. In specific, let us denote a denoiser $D(\boldsymbol{x}; \sigma)$ that minimizes the weighted denoising error for samples drawn from data distribution for all $\sigma$, i.e.,

$$\mathbb{E}_{\boldsymbol{x}_0 \sim p_{\text{data}}}\mathbb{E}_{\boldsymbol{n} \sim \mathcal{N}(\boldsymbol{0}, \sigma^2\boldsymbol{I})}\big[\lambda(\sigma)\,\|D(\boldsymbol{x}_0 + \boldsymbol{n}; \sigma) - \boldsymbol{x}_0\|_2^2\big], \quad (5)$$

then we have $\nabla_{\boldsymbol{x}}\log p(\boldsymbol{x}; \sigma) = \big(D(\boldsymbol{x}; \sigma) - \boldsymbol{x}\big)/\sigma^2$, where $\lambda(\sigma)$ is a weighting function and $\boldsymbol{n}$ is a noise. The key property of diffusion models is that the training and sampling are disentangled; one can use different sampling schemes with same denoiser.

**Text-to-image diffusion models.** Text-to-image diffusion models (Nichol et al., 2021; Saharia et al., 2022; Rombach et al., 2022) are conditional generative models that are trained with text embeddings. They utilize classifier-free guidance (CFG) (Ho & Salimans, 2022), which learns both conditional and unconditional models, and guide the sampling by interpolating the predictions with guidance scale $\omega$: $D^{\omega}(\boldsymbol{x}; \sigma, \boldsymbol{y}) = (1 + \omega)D(\boldsymbol{x}; \sigma, \boldsymbol{y}) - \omega D(\boldsymbol{x}; \sigma)$, where $\boldsymbol{y}$ is a text prompt. Empirically, $\omega > 0$ controls the tradeoff between sample fidelity and diversity. CFG scale is important in text-to-3D generation, as for the convergence of 3D optimization (Poole et al., 2022; Wang et al., 2023b).

### 2.1 TEXT-TO-3D GENERATION VIA SCORE DISTILLATION SAMPLING

Text-to-3D synthesis aims to optimize a 3D representation resembling the images generated from text-to-image diffusion models when rendered from any camera pose. This can be viewed as learning a differentiable image generator $g(\theta, c)$ where $\theta$ is a parameter for generator and $c$ is a condition for image rendering (e.g., camera pose). For 3D application, we consider optimizing Neural Radiance Fields (NeRFs) (Mildenhall et al., 2021) and 3D meshes with differentiable rasterizers (Laine et al., 2020; Munkberg et al., 2022). Throughout the paper, we omit $c$ and write $\boldsymbol{x} = g(\theta)$, unless specified.

**Score Distillation Sampling (SDS) (Poole et al., 2022).** SDS is done by differentiating the diffusion training objective (Eq. 5) with respect to the rendered image $\boldsymbol{x} = g(\theta)$. Formally, the gradient of SDS loss is given as follows:

$$\nabla_{\theta}\mathcal{L}_{\text{SDS}}\big(\theta; \boldsymbol{x} = g(\theta)\big) = \mathbb{E}_{t, \boldsymbol{n}}\left[\lambda(t)\big(\boldsymbol{x} - D_p(\boldsymbol{x} + \boldsymbol{n}; \sigma(t))\big)\frac{\partial \boldsymbol{x}}{\partial \theta}\right]. \quad (6)$$

Intuitively, the SDS perturbs the image $\boldsymbol{x}$ with noise scaled by randomly chosen timestep $t$, and guide the generator so that the rendered image moves to the higher density region. One can interpret the SDS as the gradient of probability density distillation loss (Oord et al., 2018), which is equivalent to a weighted ensemble of variational inference problem with noise scales given as follows:

$$\arg \min_\theta \mathbb{E}_{t,\boldsymbol{n}}\big[\lambda(t)D_{\mathrm{KL}}\big(q(\boldsymbol{x};\sigma(t)) \,\|\, p(\boldsymbol{x};\sigma(t))\big)\big], \text{ for } \boldsymbol{x} = g(\theta), \tag{7}$$

where we denote $q(\boldsymbol{x})$ as an implicit distribution of the rendered image $\boldsymbol{x} = g(\theta)$. It has been shown that SDS requires higher CFG scale than conventional diffusion sampling (e.g., $\omega = 100$), which often leads to blurry and over-saturated output.

**Variatioanl Score Distillation (VSD) (Wang et al., 2023b).**  To tackle the limitation of SDS, VSD solves Eq. 7 with particle-based variational inference (Chen et al., 2018) method principled by Wasserstein gradient flow (Liu, 2017). To approximate the Wasserstein gradient flow, it involves assessing the score function of rendered images $\nabla_{\boldsymbol{x}} \log q(\boldsymbol{x};\sigma)$, thus they propose to train auxiliary diffusion model for approximation. In practice, they use low-rank adapter (Hu et al., 2021) to fine-tune the diffusion model during the optimization by minimizing Eq. 5 with respect to the rendered images. Let us denote the fine-tuned diffusion model $D_q(\,\cdot\,;\sigma)$, then the gradient of VSD loss is given by

$$\nabla_\theta \mathcal{L}_{\mathrm{VSD}}\big(\theta; \boldsymbol{x} = g(\theta)\big) = \mathbb{E}_{t,\boldsymbol{n}}\left[\lambda(t)\big(D_q(\boldsymbol{x}+\boldsymbol{n};\sigma(t)) - D_p(\boldsymbol{x}+\boldsymbol{n};\sigma(t))\big)\frac{\partial \boldsymbol{x}}{\partial \theta}\right]. \tag{8}$$

Note that Eq. 8 is equivalent to Eq. 6 if we consider an ideal denoiser, i.e., $D_q(\boldsymbol{x}+\boldsymbol{n};\sigma) = \boldsymbol{x}$. In contrast to SDS, VSD allows CFG scale as low as conventional diffusion samplers (e.g., $\omega = 7.5$), thus resulting in highly-detailed and diverse 3D scene. Despite its promising quality, VSD requires a considerably long time to generate a high-quality 3D content.

## 2.2 SCHRODINGER BRIDGE PROBLEM

Schrödinger Bridges (SB) problem  (Schrödinger, 1932; Léonard, 2013), which generalizes SGM to nonlinear structure (Chen et al., 2021), aims at finding the most likely evolution of distribution between $\boldsymbol{x}_0 \sim p_A(\boldsymbol{x}_0)$ and $\boldsymbol{x}_T \sim p_B(\boldsymbol{x}_T)$ with following forward and backward SDEs:

$$\mathrm{d}\boldsymbol{x} = [\boldsymbol{f} + g^2 \nabla_{\boldsymbol{x}} \log \Phi_t(\boldsymbol{x})]\mathrm{d}t + g(t)\mathrm{d}\boldsymbol{w}, \ \ \mathrm{d}\boldsymbol{x} = [\boldsymbol{f} - g^2 \nabla_{\boldsymbol{x}} \log \hat{\Phi}_t(\boldsymbol{x})]\mathrm{d}t + g(t)\mathrm{d}\boldsymbol{w}, \quad (9,10)$$

where $\Phi_t, \hat{\Phi}_t$ are Schrödinger factors satisfying the boundary conditions $\Phi(\boldsymbol{x},0)\hat{\Phi}(\boldsymbol{x},0) = p_A(\boldsymbol{x})$ and $\Phi(\boldsymbol{x},T)\hat{\Phi}(\boldsymbol{x},T) = p_B(\boldsymbol{x})$. Eq. 9 and Eq. 10 induce the same marginal density $p_t(\boldsymbol{x})$ almost surely and the solution of evolutionary trajectory satisfies Nelson's duality (Nelson, 2020): $p_t(\boldsymbol{x}) = \Phi_t(\boldsymbol{x})\hat{\Phi}_t(\boldsymbol{x})$. Note that SGM is a special case of SB if we set $\Phi_t(\boldsymbol{x}) = 1$ and $\hat{\Phi}_t(\boldsymbol{x}) = p_t(\boldsymbol{x})$ for $t \in [0,T]$. SB is used for solving various image-to-image translation problems (Su et al., 2022; Meng et al., 2021; Liu et al., 2023).

## 3 DREAMFLOW: ELUCIDATED TEXT-TO-3D OPTIMIZATION

We present *DreamFlow*, an efficient text-to-3D optimization method by solving probability flow ODE for multi-view 3D scene. At high level, we aim at emulating the generative process of diffusion models for text-to-3D optimization, instead of utilizing diffusion training objective as in SDS. This is done by approximating the solution trajectory of probability flow ODE to transport the views of a 3D scene into the higher density region of data distribution learned by diffusion model. This, in turn, provides faster 3D optimization than SDS, while ensuring high-quality.

**Challenges in solving probability flow ODE in 3D optimization.**  However, it is not straight-forward to utilize PF ODE for text-to-3D optimization as is, because of the inherent differences between 2D sampling and 3D optimization. First, unlike 2D diffusion sampling which starts off from Gaussian noise, the multi-view images of a 3D scene are initialized by the 3D representation. As such, it is natural to cast text-to-3D optimization as a multi-view Schrödinger Bridge problem. Second, text-to-3D generation requires transporting multi-view images to the data distribution, which precludes the application of well-known image-to-image translation method (e.g., SDEdit (Meng et al., 2021)). To overcome those challenges, we first derive a PF ODE that solves SB problem (Section 3.1), and present a method that approximates probability flow for text-to-3D optimization (Section 3.2). Finally, we introduce DreamFlow, which leverages proposed optimization algorithm for efficient and high-quality text-to-3D generation (Section 3.3).

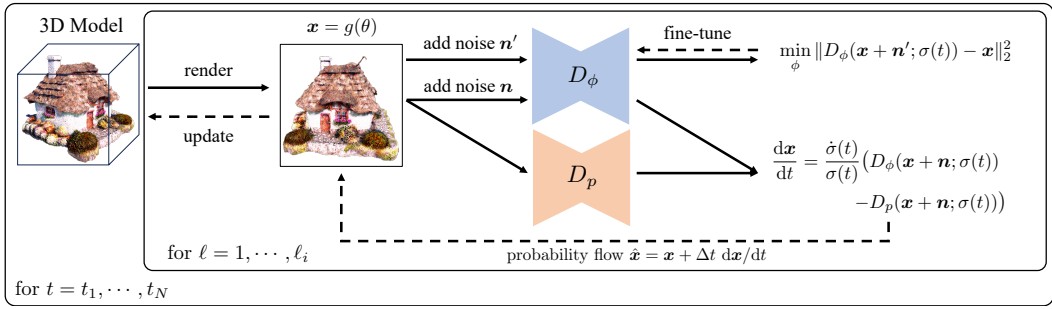

Figure 2: **Proposed 3D optimization method APFO.** APFO use predetermined timestep schedule for efficient 3D optimization. At each timestep $t_i$, we sample $\ell_i$ multi-view images from a 3D scene, and update 3D representation by approximation of probability flow computed by Eq. 13.

## 3.1 PROBABILITY FLOW ODE FOR SCHRÖDINGER BRIDGE PROBLEM

We consider the Schrödinger Bridge problem of transporting rendered image to the data distribution, while using the diffusion prior. Let $q(\boldsymbol{x})$ be the distribution of rendered image $\boldsymbol{x} = g(\theta)$. Our goal is to solve SB problem that transports $\boldsymbol{x} \sim q(\boldsymbol{x})$ to $p_{\text{data}}$ by solving the reverse SDE in Eq. 10. To leverage the rich generative prior of pre-trained diffusion model, our idea is to match the marginal density of the evolution of SB by the density $p_t$, where its score function can be estimated by pre-trained diffusion model $D_p$. Then from Nelson's duality, we have $\Phi(\boldsymbol{x}, t)\hat{\Phi}(\boldsymbol{x}, t) = p_t(\boldsymbol{x})$, where it is equivalent to $\nabla_{\boldsymbol{x}} \log \hat{\Phi}_t(\boldsymbol{x}) = \nabla_{\boldsymbol{x}} \log p_t(\boldsymbol{x}) - \nabla_{\boldsymbol{x}} \log \Phi_t(\boldsymbol{x})$. By plugging it into Eq. 10, we have our SDE and the corresponding PF ODE given as follows:

$$\mathrm{d}\boldsymbol{x} = \left[ \boldsymbol{f}(\boldsymbol{x}, t) - \frac{1}{2}g^2(t)\left( \nabla_{\boldsymbol{x}} \log p_t(\boldsymbol{x}) - \nabla_{\boldsymbol{x}} \log \Phi_t(\boldsymbol{x}) \right) \right] \mathrm{d}t. \tag{11}$$

While the score function $\nabla_{\boldsymbol{x}} \log p_t(\boldsymbol{x})$ can be estimated by pre-trained diffusion model, $\nabla_{\boldsymbol{x}} \log \Phi_t(\boldsymbol{x})$ is intractable in general. One can simply let $\Phi(\boldsymbol{x}, t) = 1$, which is indeed equivalent to SDEdit (Meng et al., 2021), but this might results in convergence to a low-density region due to its high variance as similar to SDS. Instead, we directly approximate $\nabla_{\boldsymbol{x}} \log \Phi_t(\boldsymbol{x})$ by leveraging SGM. Liu et al. (2023) showed that $\nabla_{\boldsymbol{x}} \log \Phi_t(\boldsymbol{x})$ is indeed a score function of the marginal density of forward SDE (Eq. 1) for $\boldsymbol{x} \sim q(\boldsymbol{x})$. Thus, we approximate the score $\nabla_{\boldsymbol{x}} \log \Phi_t(\boldsymbol{x})$ by learning a SGM with the sample $\boldsymbol{x} \sim q(\boldsymbol{x})$. Following (Wang et al., 2023b), this is done by fine-tuning the diffusion model $D_\phi$, initialized from $D_p$, using a low-rank adapter (Hu et al., 2021).

## 3.2 APPROXIMATE PROBABILITY FLOW ODE (APFO)

Since the forward SDE has a non-linear drift, Eq. 11 is no longer a diffusion process. Thus, we approximate the score functions $\nabla_{\boldsymbol{x}} \log p_t(\boldsymbol{x})$ and $\nabla_{\boldsymbol{x}} \log \Phi_t(\boldsymbol{x})$ by adding noise $\boldsymbol{n}$ to the sample $\boldsymbol{x}$, and use diffusion models $D_p$ and $D_\phi$, respectively. Then the Eq. 11 with diffusion models is given by

$$\mathrm{d}\boldsymbol{x} = \left[ \frac{\dot{\sigma}(t)}{\sigma(t)} \left( D_\phi(\boldsymbol{x} + \boldsymbol{n}; \sigma(t)) - D_p(\boldsymbol{x} + \boldsymbol{n}; \sigma(t)) \right) \right] \mathrm{d}t, \tag{12}$$

where we note that the same noise $\boldsymbol{n}$ is passed to diffusion models to lower the variance. Also, we use decreasing sequence of noise levels $\sigma(t)$ to ensure convergence to the data distribution.

**Amortized sampling.** The remaining challenge is to update the multi-view images within a 3D scene. To this end, we use a simple amortized sampling (Feng et al., 2017). Given an image renderer $g(\theta, c)$ with camera pose parameter $c$, we randomly sample multiple camera poses to update a whole scene at each timestep. Formally, given a decreasing noise schedule $\sigma(t)$ with timesteps $\{t_i\}_{i=1}^{N}$ (i.e., $\sigma(t_1) > \cdots > \sigma(t_N)$), we randomly sample $\ell_i$ different views at each timestep $t_i$, and update $\theta$ using the target obtained by Eq. 12. For $t = t_1, \ldots, t_N$ and for $\ell = 1, \ldots, \ell_i$, we do

$$\theta \leftarrow \arg\min_{\theta} \| g(\theta, c) - \mathbf{sg}(\boldsymbol{x} + (t_{i+1} - t_i)\boldsymbol{d}_i) \|_2^2, \quad \text{where } \boldsymbol{x} = g(\theta, \boldsymbol{c}), \text{ and}$$

$$\boldsymbol{d}_i = \frac{\sigma(t_{i+1}) - \sigma(t_i)}{\sigma(t_i)} \left( D_\Phi(\boldsymbol{x} + \boldsymbol{n}; \sigma(t_i)) - D_q(\boldsymbol{x} + \boldsymbol{n}; \sigma(t_i)) \right). \tag{13}$$

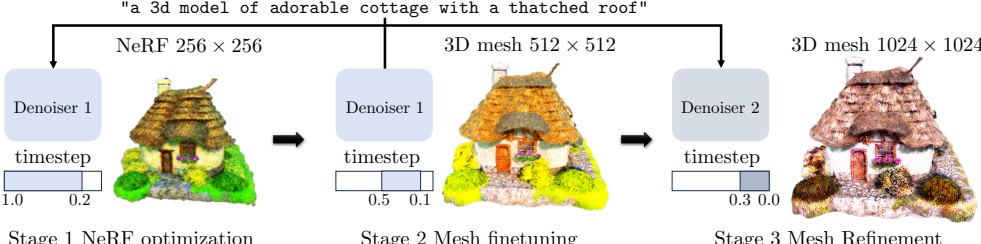

Figure 3: **Coarse-to-fine text-to-3D optimization framework of DreamFlow.** Our text-to-3D generation is done in coarse-to-fine manner; we first optimize NeRF, then extract 3D mesh and fine-tune. We use same latent diffusion model (denoiser 1) for first and second stage. Lastly, we refine 3D mesh with high-resolution latent diffusion prior (denoiser 2). At each stage, we optimize with different timestep schedule, which effectively utilize the diffusion priors.

Here sg is a stop-gradient operator. We refer our optimization method as *approximate probability flow ODE* (APFO), which we provide visual explanation in Figure 2, and the algorithm in Algorithm 1. Remark that we set different timestep schedules for different 3D optimization stages. For example, when optimizing NeRF from scratch, we start from large timestep to fully exploit the generative knowledge of diffusion model. When we fine-tune the 3D mesh, we set smaller initial timestep to add more details without changing original contents.

**Comparison with score distillation methods.** The key difference of our method and score distillation methods is their objective in leveraging generative diffusion prior. Score distillation methods directly differentiate through diffusion training loss (i.e., Eq. 5), while our approach aim at matching the density of sampling trajectory. We remark that Eq. 12 is similar to Eq. 8 as they both contain subtraction of score functions, but the difference occurs in its scaling of gradient flow. Empirically, our approach gradually minimizes the loss, while score distillation methods results in fluctuating loss (and norm of gradients) (see Figure 6). This often results in content shifting during optimization, resulting in bad geometry as well. See Appendix C for more discussion.

### 3.3 COARSE-TO-FINE TEXT-TO-3D OPTIMIZATION FRAMEWORK

We present our coarse-to-fine text-to-3D optimization framework DreamFlow, which utilizes APFO in training 3D representations. DreamFlow is consists of three stages; first we train the NeRF from scratch. Second, we extract 3D mesh from the NeRF and fine-tune. Finally, we refine 3D mesh with high-resolution diffusion prior to enhance aesthetic quality. Our framework is depicted in Figure 3.

**Stage 1: NeRF optimization.** We optimize NeRF from scratch by using multi-resolution hash grid encoder from Instant NGP (Müller et al., 2022) with MLPs attached to predict RGB colors and densities. We follow the practice from the prior works (Poole et al., 2022; Lin et al., 2023; Wang et al., 2023b) for density bias initialization, point lighting, camera augmentation, and NeRF regularization loss (see Appendix D for details). We use the latent diffusion model (Rombach et al., 2022) as a diffusion prior. During training, we render $256{\times}256$ images and use APFO with initial timestep $t_1 = 1.0$ decrease to $t_N = 0.2$ with $\ell = 5$ views. We use the total of 800 dense timesteps unless specified.

**Stage 2: 3D Mesh fine-tuning.** We convert the neural field into the Signed Distance Field (SDF) using the hash grid encoder from the first stage. Following (Chen et al., 2023), we disentangle the geometry and texture by optimizing the geometry and texture in sequence. During geometry tuning, we do not fine-tune $D_\phi$ and let $D_\phi(\boldsymbol{x}; \sigma) = \boldsymbol{x}$ as done in ProlificDreamer (Wang et al., 2023b). For geometry tuning, we use timesteps from $t_1 = 0.8$ to $t_N = 0.4$, and for texture tuning, we use timesteps from $t_1 = 0.5$ to $t_N = 0.1$. We all use $\ell = 5$.

**Stage 3: Mesh refinement.** We refine the 3D mesh using high-resolution diffusion prior, SDXL refiner (Podell et al., 2023), which is trained to enhance the aesthetic quality of an image. During refinement, we render the image with resolution of $1024{\times}1024$, and use $t_1 = 0.3$ to $t_N = 0.0$ with $\ell = 10$. Here we use timestep spacing of 10, which suffices to enhance the quality of 3D mesh.

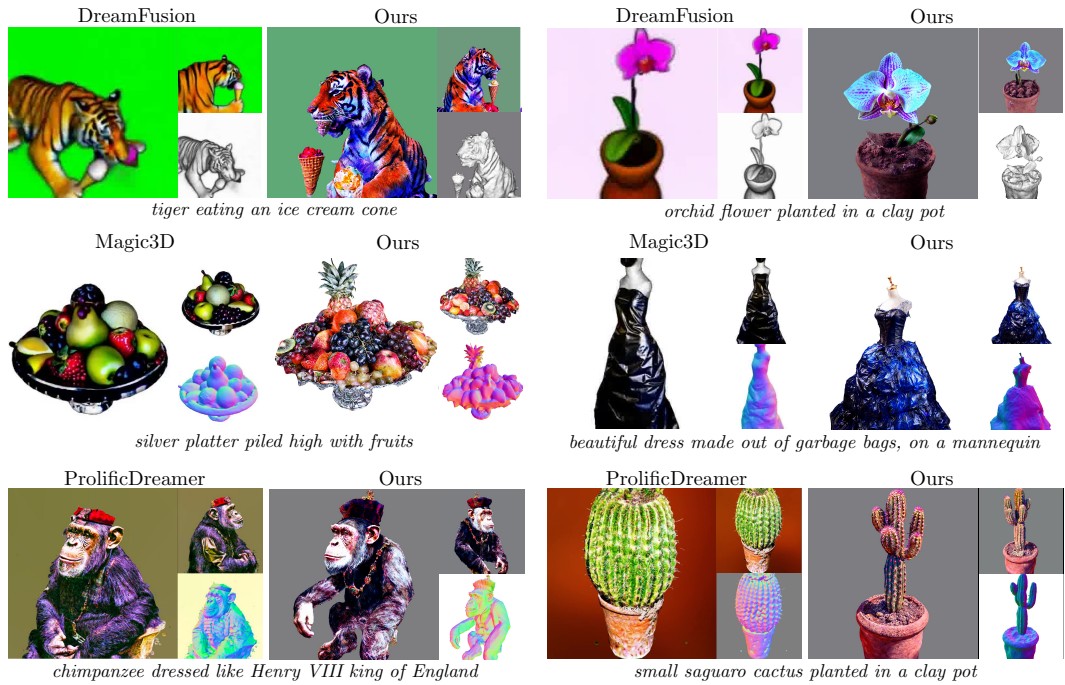

| DreamFusion | Ours | DreamFusion | Ours |

*tiger eating an ice cream cone*          *orchid flower planted in a clay pot*

*silver platter piled high with fruits*          *beautiful dress made out of garbage bags, on a mannequin*

*chimpanzee dressed like Henry VIII king of England*          *small saguaro cactus planted in a clay pot*

Figure 4: **Qualitative comparison with baseline methods.** For each baseline method, we present visual examples with same text prompt is given. Our approach presents more detailed textures.

## 4 EXPERIMENT

Throughout the experiments, we use the text prompts in the DreamFusion gallery[2] to compare our method with the baseline methods DreamFusion (Poole et al., 2022), Magic3D (Lin et al., 2023), and ProlificDreamer (Wang et al., 2023b).

**Qualitative comparisons.** In Figure 4, we present qualitative comparisons between the baseline methods. We take baseline results from figures or videos in respective papers or websites. Compared to SDS-based approach such as DreamFusion and Magic3D, our approach presents more photorealistic details and shapes. When compared to ProlificDreamer, our approach results in more photorealistic details (e.g., eyes are more clearly represented for chimpanzee, and shadows are more clearly depicted in cactus) by exploiting high-resolution diffusion prior.

**User preference study.** We conduct user studies to measure the human preference compared to baseline methods. For each baseline method, we select 20 3D models from their original papers or demo websites (see Appendix A). Then we construct three binary comparison tasks (total 60 comparison) to rank between DreamFlow and each baseline method. For each pair, 10 human annotators were asked to rank two videos (or images if videos were unavailable) based on following three components: text prompt fidelity, 3D consistency, and photorealism (see Appendix D). Results are in Table 1. Compared to DreamFusion, Magic3D, and ProlificDreamer, DreamFlow consistently wins on photorealism. While DreamFusion remains better, ours shows on par or better performance against more recent methods, Magic3D and ProlificDreamer, on 3D consistency and prompt fidelity.

**Quantitative comparison.** For quantitative evaluation, we measure the CLIP (Radford et al., 2021) R-precision following the practice of DreamFusion (Poole et al., 2022). We compare with Dream-Fusion and ProlificDreamer on NeRF generation, and Magic3D and ProlificDreamer for 3D mesh fine-tuning. For NeRF generation comparison, we randomly select 100 text prompts from DreamFusion gallery and reproduce results for ProlificDreamer. For DreamFusion, we simply take videos from their gallery. For 3D mesh fine-tuning comparison, we select 50 NeRF representations from previous

---

[2]https://dreamfusion3d.github.io/gallery.html

Table 1: **User preference studies.** We conduct user studies to measure the preference for 3D models. We present pairwise comparison between DreamFlow (ours) and baselines DreamFusion (DF) (Poole et al., 2022), Magic3D (M3D) (Lin et al., 2023), and ProlificDreamer (PD) (Wang et al., 2023b).

| Method | ours | tie | DF | ours | tie | M3D | ours | tie | PD |
|---|---|---|---|---|---|---|---|---|---|
| 3D consistency | 26.6% | 31.0% | **42.4%** | 36.9% | **51.8%** | 11.3% | 31.9% | **52.5%** | 15.6% |
| Prompt fidelity | 22.8% | **43.5%** | 33.7% | 38.1% | **52.4%** | 9.52% | 30.6% | **58.8%** | 10.6% |
| Photorealism | **82.1%** | 6.52% | 11.4% | **86.3%** | 8.93% | 4.76% | **43.8%** | 36.9% | 19.4% |

Table 2: **Quantitative comparisons: NeRF optimization.** We measure average CLIP R-precision scores of rendered views from DreamFusion, ProlificDreamer and DreamFlow.

| Method | ViT-L/14 | ViT-B/16 | ViT-B/32 |
|---|---|---|---|
| DreamFusion | 0.846 | 0.796 | 0.706 |
| ProlificDreamer | 0.875 | 0.858 | 0.782 |
| DreamFlow (ours) | **0.905** | **0.871** | **0.798** |

Table 3: **Quantitative comparisons: Mesh fine-tuning.** We measure CLIP R-precision of rendered 3D mesh from Magic3D, ProlificDreamer, and DreamFlow.

| Method | ViT-L/14 | ViT-B/16 | ViT-B/32 |
|---|---|---|---|
| Magic3D | 0.801 | 0.741 | 0.599 |
| ProlificDreamer | 0.884 | 0.875 | 0.828 |
| DreamFlow (ours) | **0.902** | **0.886** | **0.846** |

experiment, and reproduce Magic3D and ProlificDreamer using open-source implementation.[3] We do not conduct mesh refinement for fair comparison. For evaluation, we render 120 views with uniform azimuth angle, and compute CLIP R-precision using various CLIP models (ViT-L/14, ViT-B/16, and ViT-B/32). Table 2 and Table 3 presents the results, which show that our method achieves better quantitative score compared to prior methods.

**Optimization efficiency.** We show the efficiency of our method through comparing the generation speed. Note that we use a single A100 GPU for generating each 3D content. For the NeRF optimization, we train for 4000 iterations with resolution of 256, which takes 50 minutes. For mesh fine-tuning, we tune the geometry for 5000 iterations and texture for 2000, taking 40 minutes in total. Lastly, we refine mesh for 300 iterations, which takes 20 minutes. In sum, our method takes about 2 hours for a single 3D content generation. In comparison, DreamFusion takes 1.5 hours using 4 TPU v4 chips in generating NeRF with resolution of 64, and Magic3D takes 40 minutes using 8 A100 GPUs in synthesizing 3D mesh with resolution of 512. ProlificDreamer requires 5 hours in generating NeRF, 7 hours in mesh tuning. While Magic3D and DreamFusion could be faster using larger computing power, DreamFlow generates more photorealistic and high-resolution 3D content.

**Optimization analysis.** We compare the optimization processes of APFO and VSD by comparing the values of loss and gradient norm per optimization iteration. For VSD, we compute loss as in Eq. 13, which is equivalent to Eq. 8 in effective gradient. In Figure 6, we plot the loss values and gradient norms during NeRF optimization with each APFO and VSD. Remark that the loss and gradient norm gradually decrease for APFO, while they fluctuates for VSD. This is because of the randomly drawn timestep during optimization, which makes optimization unstable. Thus, we observe that VSD often results in poor geometry during optimization, or over-saturated (see Figure 16 in Appendix C).

**Effect of each optimization stage.** In Figure 5, we provide qualitative examples on the effect of coasre-to-fine optimization. While DreamFlow generates high-quality NeRF, mesh tuning and refinement enhances the fidelity by increasing the resolution and using larger diffusion prior.

## 5 RELATED WORK

**Text-to-3D generation.** Recent works have demonstrated the promise of generating 3D content using large-scale pre-trained text-to-image generative models without using any 3D dataset. Earlier works used CLIP (Radford et al., 2021) image-text models for aligning 2D renderings and text prompts (Jain et al., 2022; Mohammad Khalid et al., 2022). DreamFusion (Poole et al., 2022) present

---

[3] https://github.com/threestudio-project/threestudio

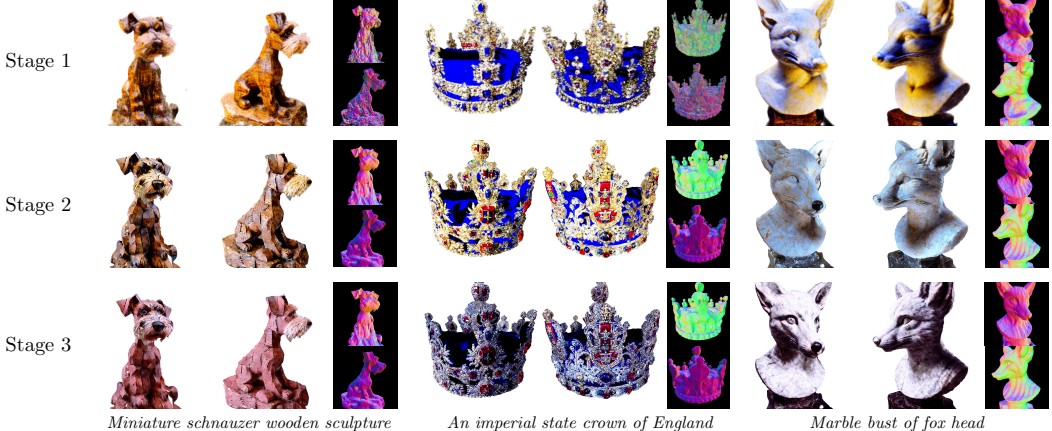

Figure 5: **Ablation on the effect of DreamFlow coarse-to-fine text-to-3D optimization.** Given stage 1 generates high quality NeRF, stage 2 improves the geometry and texture, and stage 3 refines the 3D mesh to add more photorealistic details.

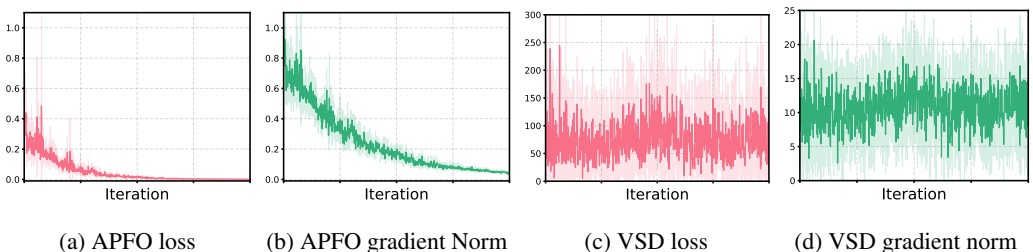

| (a) APFO loss | (b) APFO gradient Norm | (c) VSD loss | (d) VSD gradient norm |

Figure 6: **Optimization analysis.** We plot the loss and gradient norm during 3D optimization for APFO and VSD. Remark that the loss and gradient norm gradually decreases for APFO, while VSD shows fluctuating loss and gradient norm.

text-to-3D generation by distilling score function of text-to-image diffusion model. Concurrently, Score Jacobian Chaining (SJC) (Wang et al., 2023a) established another approach to use 2D diffusion prior for 3D generation. Magic3D (Lin et al., 2023) extends DreamFusion to higher resolution 3D content creation by fine-tuning on 3D meshes. Other works (Chen et al., 2023; Tsalicoglou et al., 2023) focused on designing better geometry prior for text-to-3D generation. ProlificDreamer (Wang et al., 2023b) enabled high-quality 3D generation by using advanced score distillation objective. All the prior works resort to score distillation methods, while we first demonstrate probability flow ODE based approach to accelerate the 3D optimization.

**Transferring 2D diffusion priors.** Due to the success of score-based generative models (Song et al., 2020b), especially diffusion models (Ho et al., 2020) have led to the success on text-to-image synthesis (Nichol et al., 2021; Saharia et al., 2022; Rombach et al., 2022; Balaji et al., 2022). Many recent studies focus on transferring the rich generative prior of text-to-image diffusion models to generate or manipulate various visual contents such as images, 3D representations, videos, with text prompts (Brooks et al., 2023; Haque et al., 2023; Kim et al., 2023; Hertz et al., 2023). Our approach also shares the objective in exploiting the diffusion models, where utilizing APFO to synthesize various visual contents is an interesting future work.

## 6    CONCLUSION

We propose DreamFlow, which enables high-quality and fast text-to-3D content creation. Our approach is built upon elucidated optimization strategy which approximates the probability flow ODE of diffusion generative models catered for 3D scene optimization. As a result, it significantly streamlines the 3D scene optimization, making it more scalable with high-resolution diffusion priors.

By taking this benefit, we propose three stage 3D scene optimization framework, where we train NeRF from the scratch, fine-tune the mesh extracted from NeRF, and refine the 3D mesh using high-resolution diffusion prior. Through user preference studies and qualitative comparisons, we show that DreamFlow outperforms prior state-of-the-art method, while being 5x faster in its generation.

**Limitation.** Since we are using pre-trained diffusion priors that do not have 3D understanding, the results may not be satisfactory in some cases. Also, the unwanted bias of pre-trained diffusion model might be inherited.

## ACKNOWLEDGEMENT

This work was supported by Institute for Information & communications Technology Promotion (IITP) grant funded by the Korea government (MSIT) (No.2019-0-00075 Artificial Intelligence Graduate School Program(KAIST); No.2021-0-02068, Artificial Intelligence Innovation Hub; No.2022-0-00959, Few-shot Learning of Causal Inference in Vision and Language for Decision Making). This work is in partly supported by Google Research grant and Google Cloud Research Credits program.

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

# Appendix

## A  ADDITIONAL QUALITATIVE COMPARISON

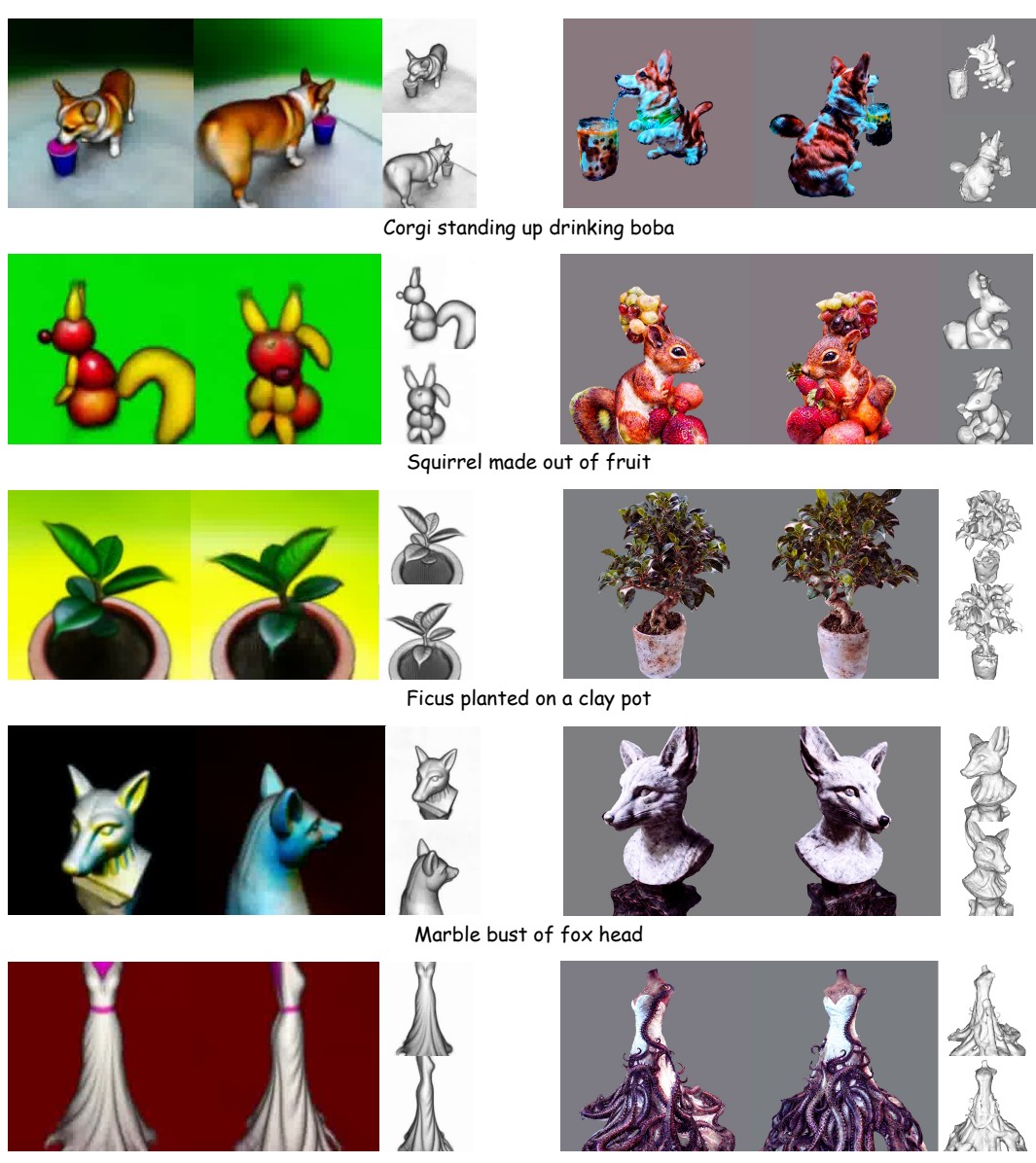

Corgi standing up drinking boba

Squirrel made out of fruit

Ficus planted on a clay pot

Marble bust of fox head

Wedding dress made out of tentacles

Figure 7: **Qualitative Comparison with DreamFusion (Poole et al., 2022) (left) and DreamFlow (right).**

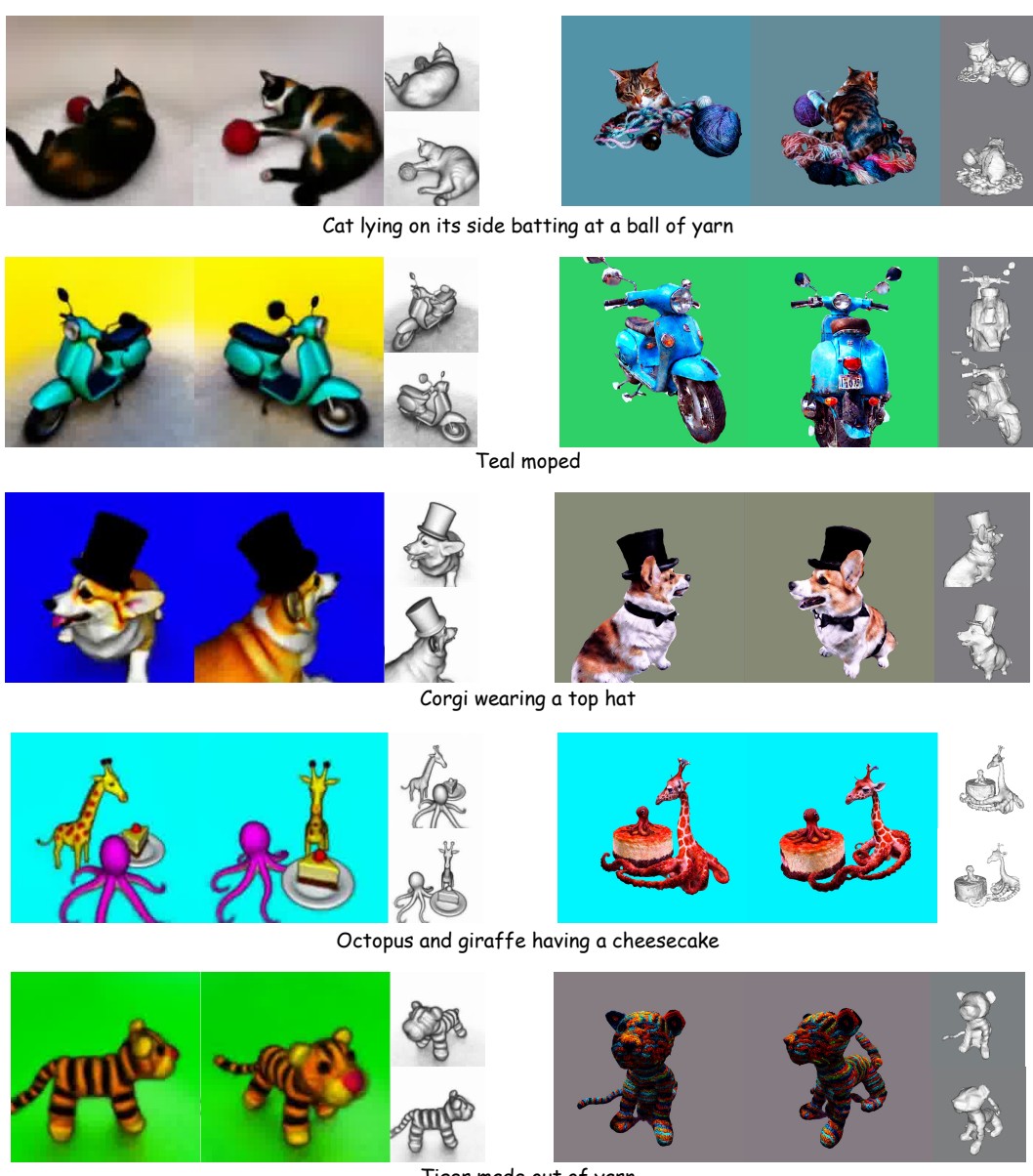

Cat lying on its side batting at a ball of yarn

Teal moped

Corgi wearing a top hat

Octopus and giraffe having a cheesecake

Tiger made out of yarn

Figure 8: **Qualitative Comparison with DreamFusion (Poole et al., 2022) (left) and DreamFlow (right).**

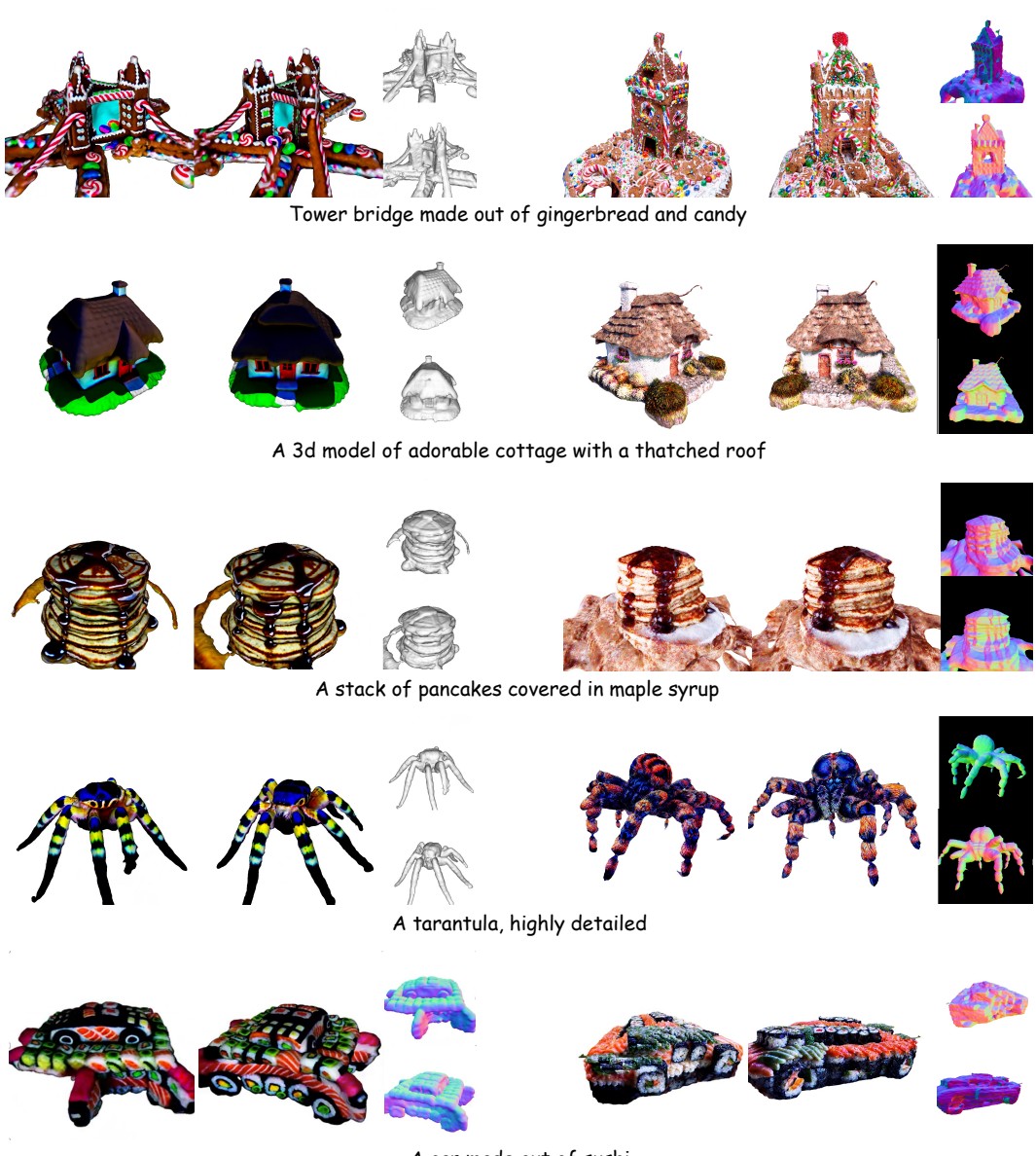

Tower bridge made out of gingerbread and candy

A 3d model of adorable cottage with a thatched roof

A stack of pancakes covered in maple syrup

A tarantula, highly detailed

A car made out of sushi

Figure 9: **Qualitative Comparison with Magic3D (Lin et al., 2023) (left) and DreamFlow (right).**

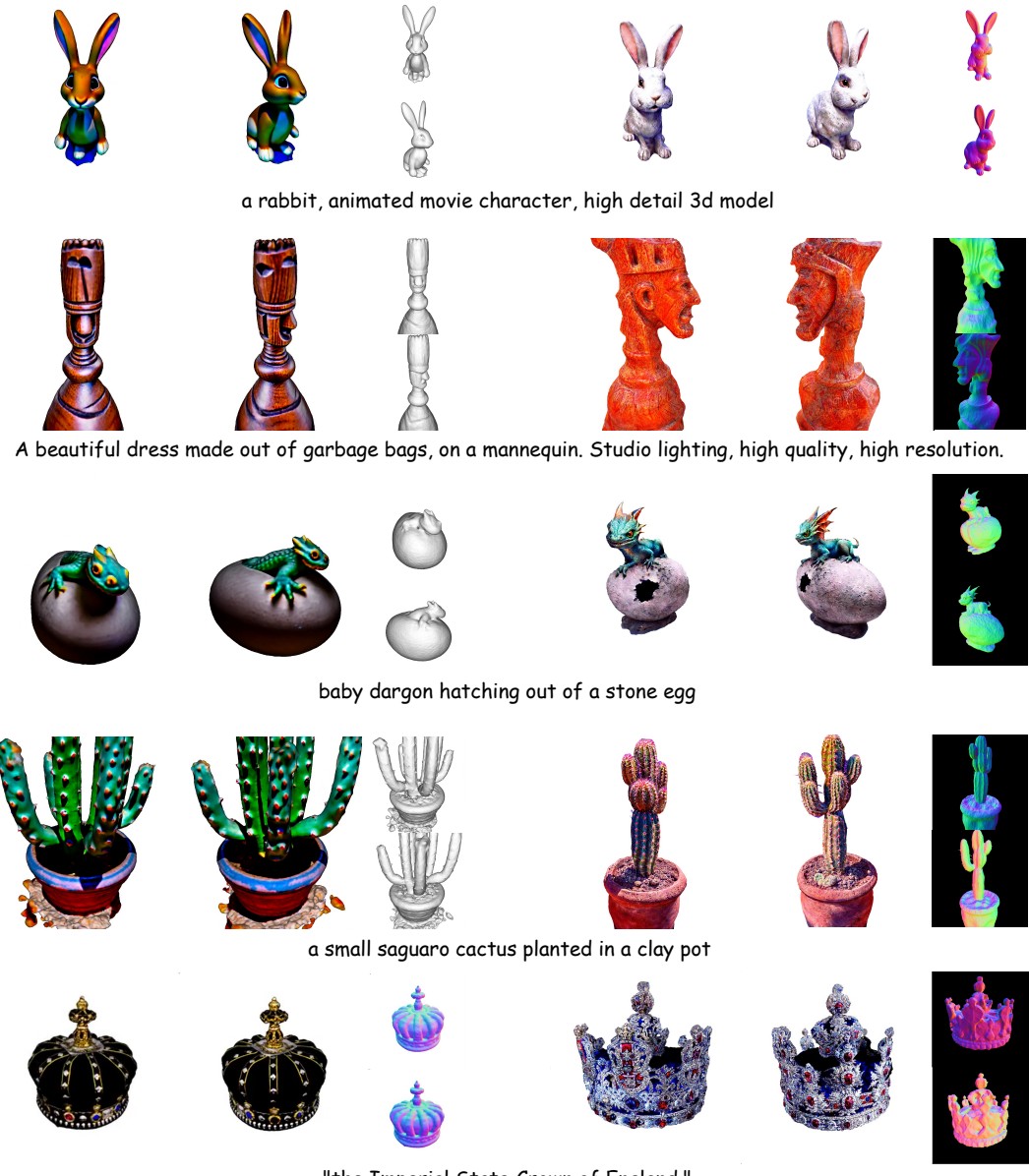

a rabbit, animated movie character, high detail 3d model

A beautiful dress made out of garbage bags, on a mannequin. Studio lighting, high quality, high resolution.

baby dargon hatching out of a stone egg

a small saguaro cactus planted in a clay pot

"the Imperial State Crown of England."

Figure 10: **Qualitative Comparison with Magic3D (Lin et al., 2023) (left) and DreamFlow (right).**

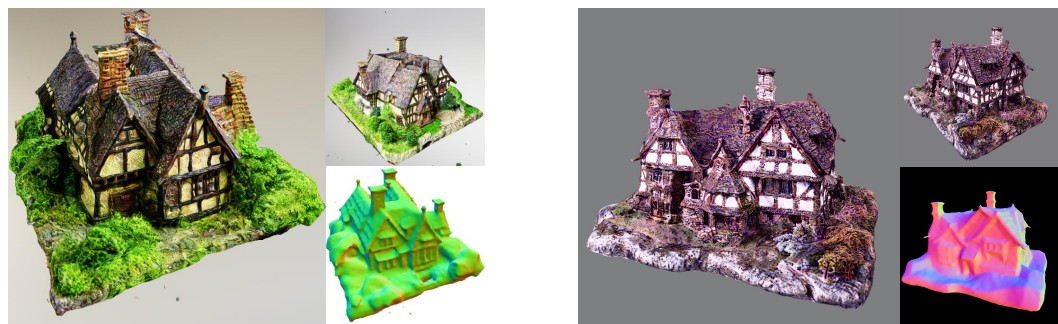

A model of a house in Tudor style

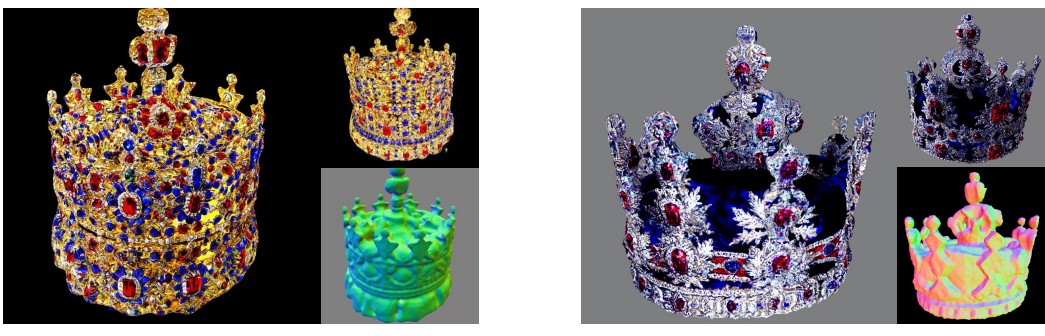

An imperial state crown of England

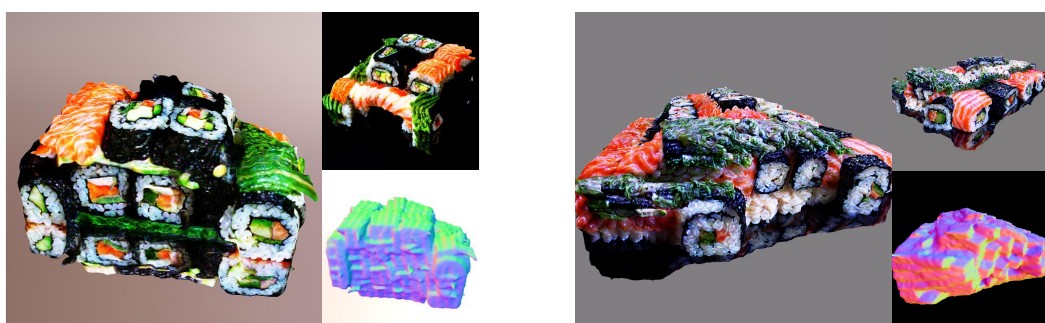

A car made out of sushi

Figure 11: **Qualitative Comparison with ProlificDreamer (Wang et al., 2023b) (left) and Dream-Flow (right).**

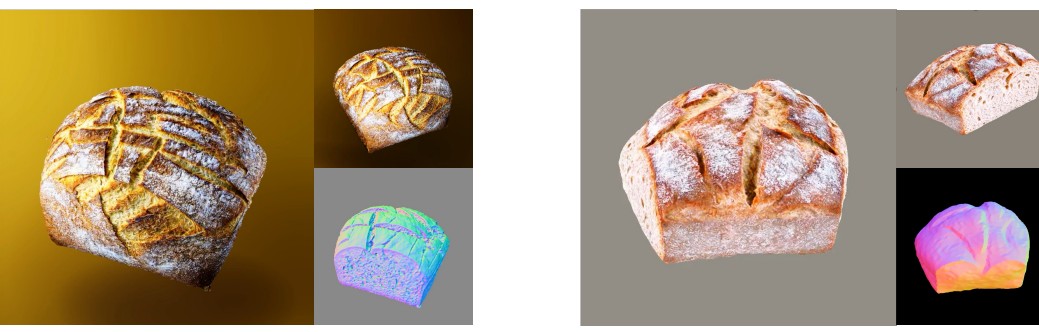

A sliced loaf of fresh bread

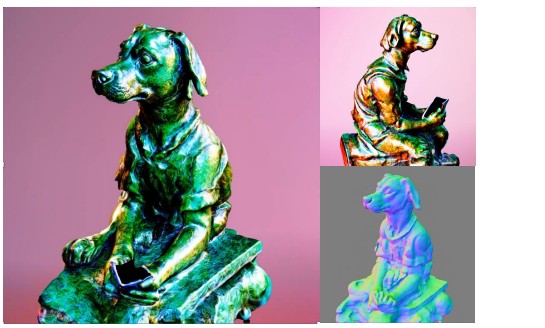 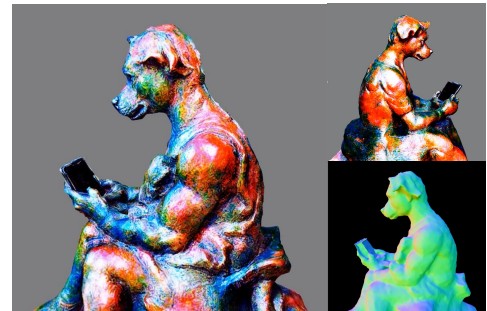

Michelangelo style statue of a dog reading news on a cell phone

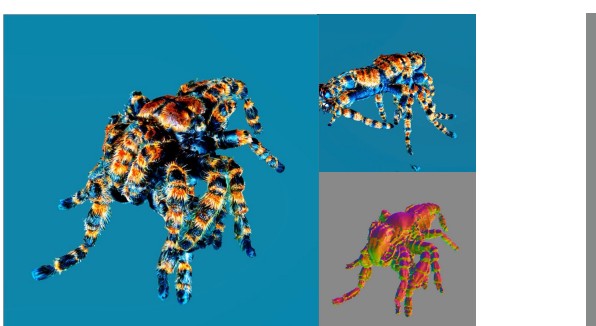 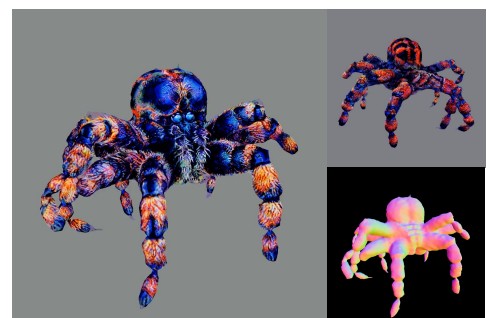

A tarantula, highly detailed

Figure 12: **Qualitative Comparison with ProlificDreamer (Wang et al., 2023b) (left) and Dream-Flow (right).**

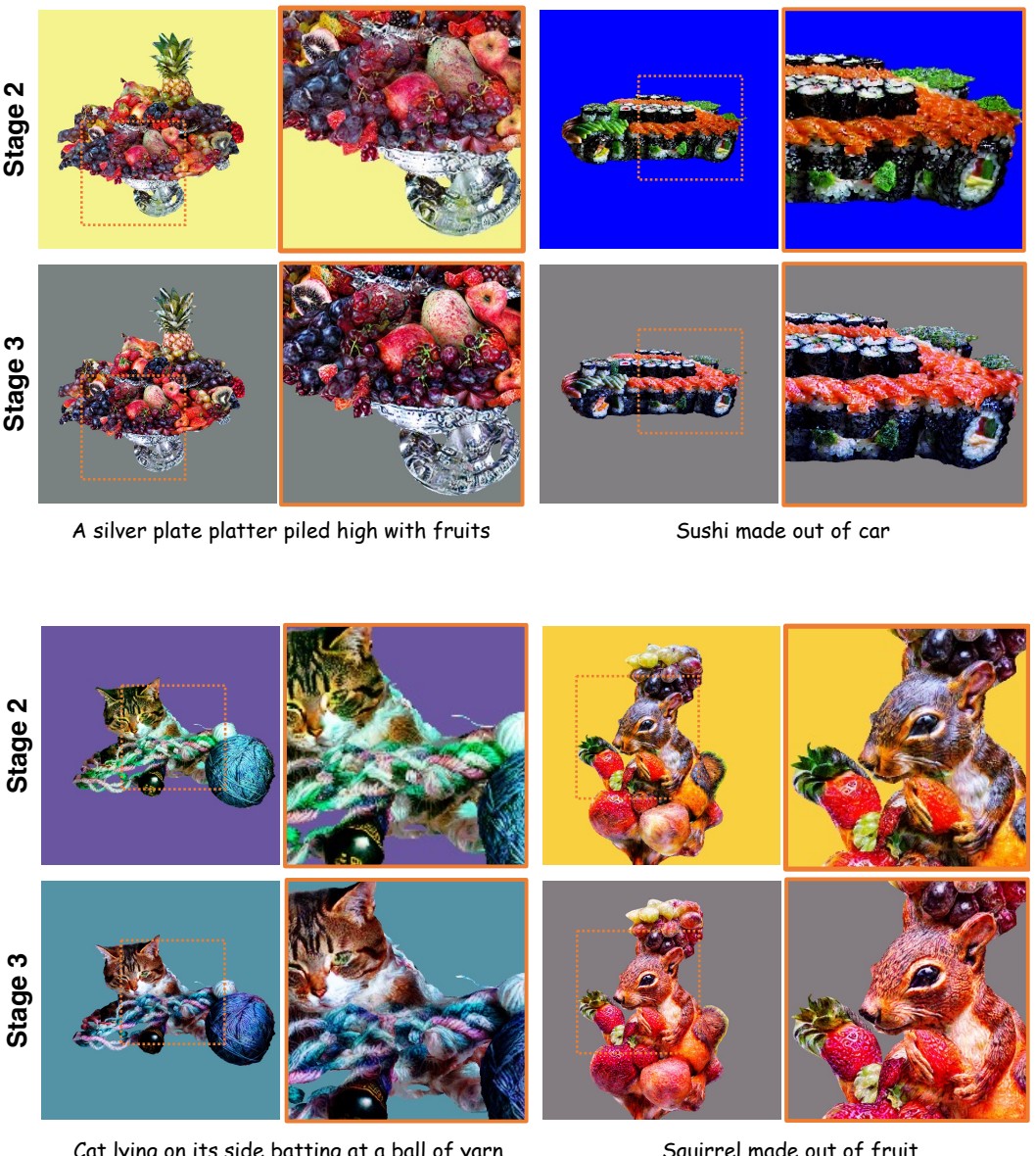

Figure 13: **Qualitative comparisons Stage 2 and Stage 3 mesh fine-tuning.** We provide additional qualitative results on the ablation of using Stage 3 Mesh refinement. We zoomed in a region of rendered view to demonstrate the effect of stage 3 mesh refinement. Remark that Stage 3 mesh results in more high-contrast images with better textures.

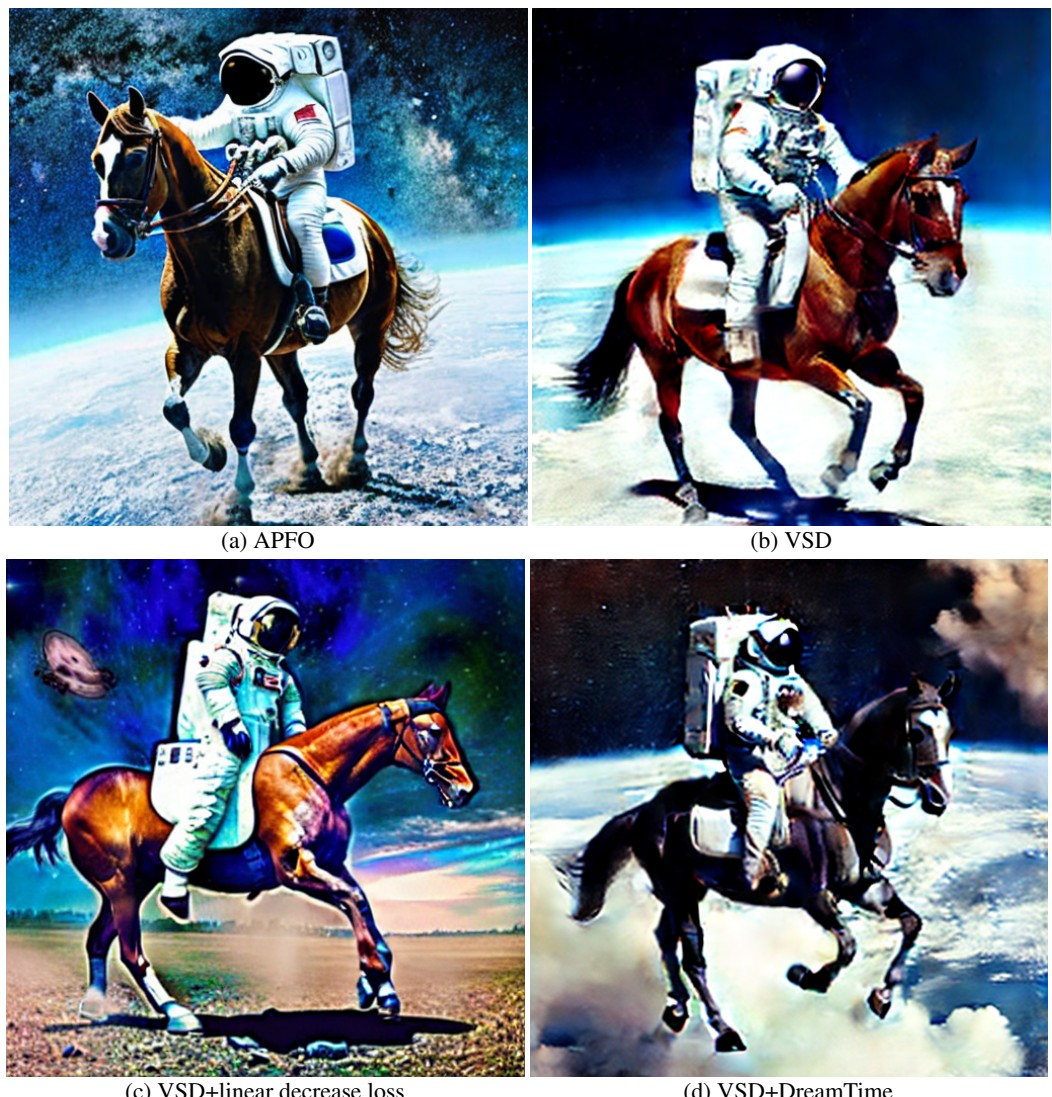

(a) APFO                                        (b) VSD

(c) VSD+linear decrease loss                    (d) VSD+DreamTime

Figure 14: **2D experiments.** We provide qualitative results of the 2D image generation with (a) APFO, (b) VSD, (c) VSD with linearly decreasing timestep schedule, and (d) VSD with DreamTime (Huang et al., 2023). Remark that APFO generates high-fidelity image with only 200 optimization steps, while VSD and its variants generate image with blurry artifacts.

## B FURTHER EXPLANATION

**Difference between APFO and VSD.** Here we provide further details in difference between APFO and VSD. First, remark that the proposed method APFO, is different to VSD in its implementation. In specific, the major difference is that APFO compute the time-derivative of the noise scale, i.e. $\frac{\dot{\sigma}(t)}{\sigma(t)}$ (e.g. Eq. 11), which is designed to transport the diffusion latent to be the next noise level from the derivation of probability flow ODE. On the other hand, score distillation methods, e.g. VSD, leverages the empirically designed weighting function $\lambda(t) = \sigma^2(t)$. Those design choices of APFO and VSD are different due to their intrinsic motivation. APFO aims to approximate the probability flow ODE with decreasing timestep schedule, so that at the terminal step (i.e., as $\sigma(t) \to 0$, the optimized images are sampled from data distribution. On the other hand, VSD aims to minimize the ensemble of KL divergence on various noise levels, where particle-based variational inference algorithms were used.

Moreover, APFO is different from VSD with a simple timestep annealing method.This is because APFO employs time-derivative of noise level, which is the exact amount to move to the next noise

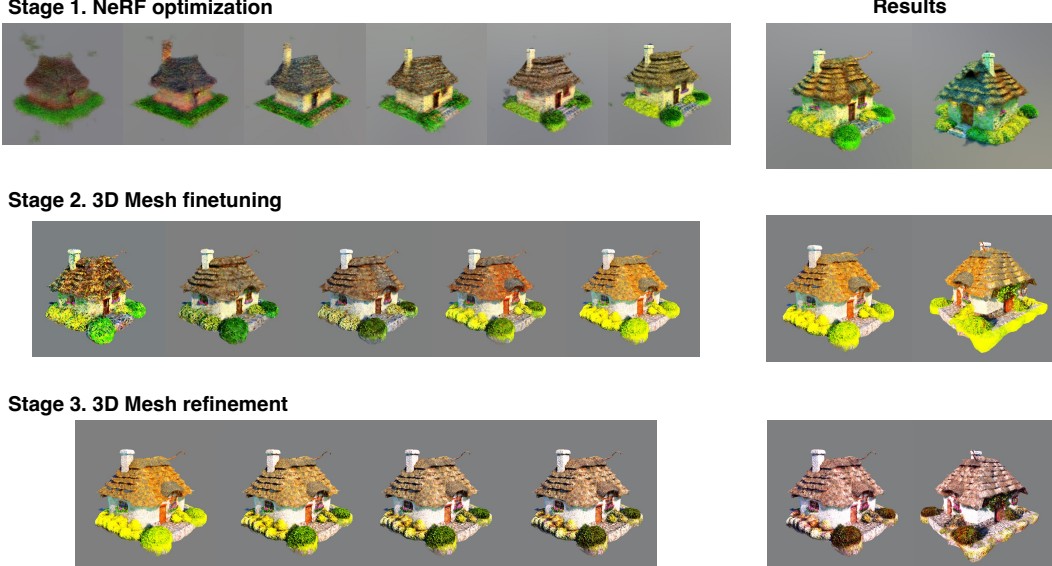

Figure 15: **Qualitative visualization text-to-3D optimization process of DreamFlow**. We show that DreamFlow generates fast and high-quality 3D representations from text prompt by using pre-scheduled timesteps.

level. However, implementing VSD with decreasing noise-level does not involve such design choice. For better explanation, we provide an 2D image generation experiment which compares APFO with default VSD (i.e. random timestep sampling) and VSD with timestep annealing (linearly decreasing as in APFO). Also, we provide additional comparison with DreamTime (Huang et al., 2023), which proposed to sample timesteps according to predetermined weighting functions. For implementation, we use 200 optimization steps by using default DDIM (Song et al., 2020a) timestep schedule. For other methods, we follow the experimental setup in ProlificDreamer (Wang et al., 2023b) paper, with 500 optimization steps. For VSD with linearly decreasing timestep, we linearly decrease timestep simply from 1.0 to 0.0, and for VSD with DreamTime, we follow their implementation of weighting function.

The results are shown in Figure 14. One can observe that the image generated by APFO results in more sharp details, while other images remain blurry artifacts. Also, the blurry artifact does not diminish even when using an annealing timestep schedule for VSD or VSD scheduled with DreamTime. Also, APFO generates a more faithful image even when using 100 or 200 number of steps, while other methods use 500 steps. This 2D experiment show that 1) APFO is not only different from simply taking VSD with timestep annealing, but also presents a more faithful image, and 2) by approximating the probability flow of pretrained diffusion model, APFO is able to generate image with fewer optimization steps.

## C   ABLATION STUDY

**Effect of predetermined timestep schedule.**   To further demonstrate the effect of predetermined timestep schedule, we plot the optimization processes of DreamFlow framework. In Figure 15, we visualize the interim results of NeRF and meshes during 3D optimization. We remark that our framework using scheduled timesteps are similar to the ancestral sampling of text-to-image diffusion models that we first identify the geometry with high noise scales (i.e., large timestep), then we provide more details by using low noise scales (i.e., small timesteps). The refinement stage is involved to enhance the quality of 3D meshes, which is a common practice in 2D image sampling, where image-to-image diffusion models are used. While our method show how we emulate the 2D text-to-image sampling procedure for text-to-3D generation, some other applications in 2D spaces can be further considered, e.g., instruction-guided editing (Brooks et al., 2023), personalization (Ruiz

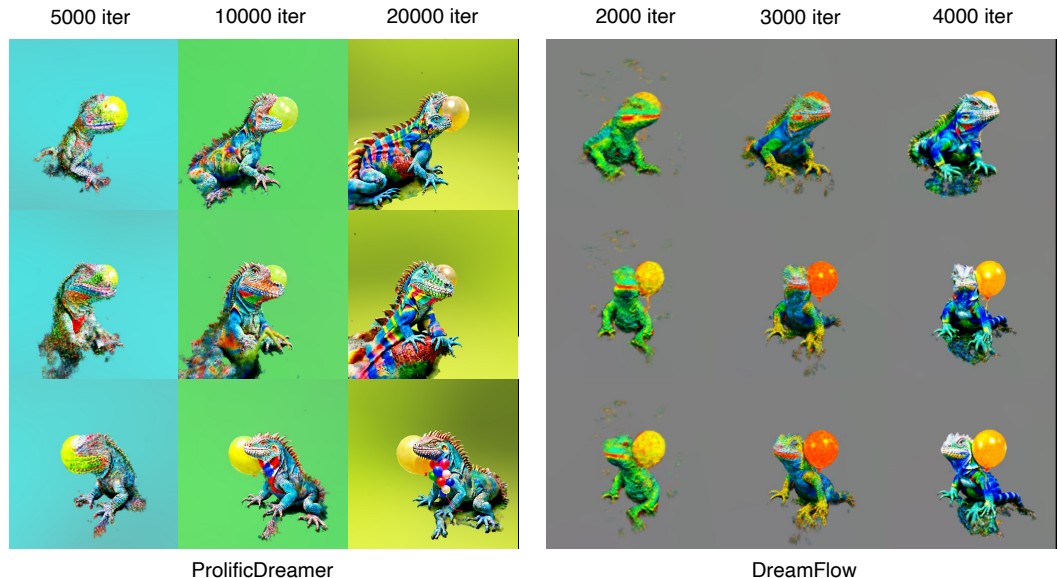

Figure 16: **Qualitative comparison between VSD and APFO optimization process.** Due to the random timestep, VSD often changes the object to bad geometry, while APFO consistently move to the data distribution. Thus is more efficient and results in better 3D generation.

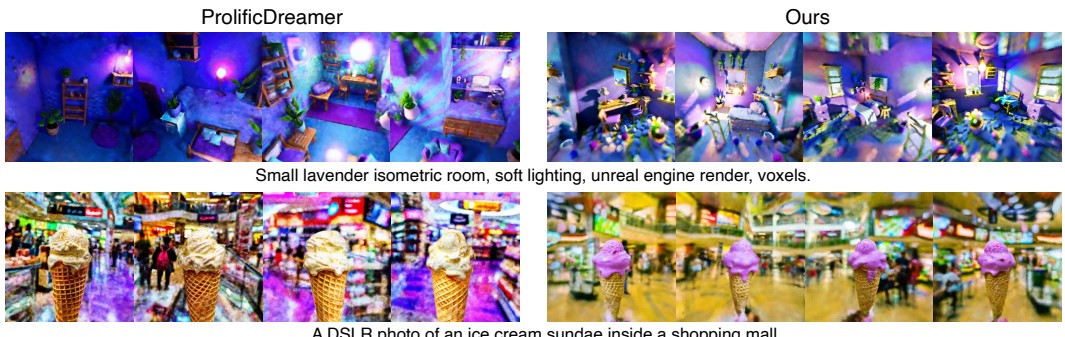

Figure 17: **Complex NeRF generation.** We compare ProlificDreamer on scene (first row) and object-centric scene (second row) generation. Our method generates higher fidelity NeRF scene, while being more faster in its generation.

et al., 2023; Gal et al., 2022), or reference-guided stylization (Sohn et al., 2023), which we leave for future work.

**Qualitative comparison on 3D optimization methods.** As shown in Figure 6, we show that APFO gradually decreases the loss and gradient norm during its optimization, while VSD suffers from fluctuating loss. We remark that using random timestep in score distillation (i.e., VSD) inefficiently prolongs the optimization, while often results in content shift during optimization, which ultimately results in bad geometry. Here, we provide qualitative examples in such cases, where we show how APFO mitigates such issue by using decreasing timestep schedule. In Figure 16, we depict the optimization process of VSD and APFO. First, note that even though APFO trained shorter than VSD, APFO converged to a valid 3D content. While length optimization of VSD leads to detailed texture, the geometry is tilted and the undesirable features are appended throughout the optimization. On the other hand, APFO consistently update the feature with decreasing noise scales, which lead to coherent geometry and texture.

**Complex NeRF generation.**    Following the setup from ProlificDreamer (Wang et al., 2023b), we evaluate our method in complex and object-centric scene generation. This is done by using their NeRF scene-initialization by using different spatial density bias. In Figure 17, we show the results compared to ProlificDreamer. Note that our approach generates scene with higher fidelity even with shorter optimization iteration.

---

**Algorithm 1** Approximate probability flow ODE (APFO)

1: **procedure** APFO($g$, $D_p(\boldsymbol{x}; \sigma)$, $D_\phi(\boldsymbol{x}; \sigma)$, $\sigma(t)$, $s(t)$, $t_{i \in \{0,\dots,N\}}$, $\ell_{i \in \{0,\dots,N\}}$)
2:     **Initialize** $\theta$                                                                   ▷ 3D scene initialization
3:     **for** $i \in \{0,\dots,N-1\}$ **do**
4:         **for** $j \in \{0,\dots,\ell_i\}$ **do**                                ▷ Sample $\ell_i$ views for amortized optimization
5:             Sample $c_j \sim \mathcal{C}$ and render $\boldsymbol{x} = g(\theta, c_j)$
6:             Sample $\boldsymbol{n} \sim \mathcal{N}\big(\mathbf{0},\ \sigma^2(t_i)\mathbf{I}\big)$
7:             $\phi \leftarrow \phi - \eta_\phi \nabla_\phi \|D_\phi(\boldsymbol{x} + \boldsymbol{n}; \sigma(t_i)) - \boldsymbol{x}\|_2^2$          ▷ Update $\phi$ using DSM objective
8:             Sample $\boldsymbol{n}' \sim \mathcal{N}\big(\mathbf{0},\ \sigma^2(t_i)\mathbf{I}\big)$                     ▷ Sample another random noise
9:             $\boldsymbol{d} \leftarrow \dfrac{\dot{\sigma}(t_i)}{\sigma(t_i)}\big(D_p(\boldsymbol{x} + \boldsymbol{n}'; \sigma(t_i)) - D_\phi(\boldsymbol{x} + \boldsymbol{n}'; \sigma(t_i))\big)$       ▷ Evaluate $\mathrm{d}\boldsymbol{x}/\mathrm{d}t$ at $t_i$
10:            $\hat{\boldsymbol{x}} \leftarrow \boldsymbol{x} + (t_{i+1} - t_i)\boldsymbol{d}$                        ▷ Take Euler step from $t_i$ to $t_{i+1}$
11:            $\theta \leftarrow \theta - \eta_2 \nabla_\theta \|g(\theta, c_j) - \hat{\boldsymbol{x}}\|_2^2$                  ▷ Update $\theta$ using updated target
12:     **return** $\theta$                                                           ▷ Return optimized 3D scene

---

# D  EXPERIMENTAL SETUP

## D.1  3D SCENE OPTIMIZATION

**Diffusion model configuration.** We use Stable Diffusion 2.1 (Rombach et al., 2022) for our latent diffusion prior of stage 1 and stage 2, and Stable Diffusion XL (SDXL) Refiner (Podell et al., 2023) for stage 3. When estimating $\nabla_{\boldsymbol{x}} \log \Phi_t(\boldsymbol{x})$, we use Stable Diffusion 2.1-v, a v-prediction model (Salimans & Ho, 2022) for stage 1 and stage 2, and use same SDXL Refiner for stage 3. Following (Wang et al., 2023b), we train auxiliary camera pose embedding MLP, which is added to timestep embedding for each U-Net block. However, we find not training camera embedding also works well. We use `EulerDiscreteScheduler` for our noise schedule and timestep.

**Scene representation.** For density bias initialization, we follow Magic3D (Lin et al., 2023) and ProlificDreamer (Wang et al., 2023b). We set the camera distance from 1.0 to 1.5, bounding box size as 1.0. We use `softplus` activation for the density prediction and add spatial density given by $\tau_{\text{init}}(\mu) = \lambda_\tau \cdot (1 - \|\mu\|_2 / c)$, where $\mu$ is a 3D coordinate, $\lambda_\tau = 10$ is a density bias scale, $c = 0.5$ is the offset scale. We use same light sampling strategy as of Magic3D. We use background MLP that learns background color following (Poole et al., 2022).

**NeRF optimization.** We use hash grid encoder from Instant NGP (Müller et al., 2022), with 16 levels of hash dictionaries of size $2^{19}$ and feature dimension of 4. We use 512 samples per ray in rendering. During NeRF optimization, we found that orientation loss (Verbin et al., 2022) helps consolidating the geometry. We do not use sparsity regularization, which we find it hurts geometry. During optimization, we use AdamW optimizer (Loshchilov & Hutter, 2017) where we train the grid encoder with learning rate $1e - 2$, color and density network with learning rate $1e - 3$, and background MLP with learning rate $1e - 3$ or $1e - 4$. We do not use shading, because it distracts learning texture.

**Stage 2: 3D mesh fine-tuning.** We extract 3D mesh from stage 1 NeRF using DMTet (Shen et al., 2021). We then fine-tune the geometry by rendering normal maps (Chen et al., 2023). We find that finetuning $D_\phi$ with APFO for geometry tuning is not that useful, thus we simply let $D_\phi(\boldsymbol{x}; \sigma) = \boldsymbol{x}$ for given rendered image, and use same APFO algorithm. We fine-tune the geometry with AdamW optimizer of learning rate $1e - 2$. For texture fine-tuning, we use AdamW optimizer with learning rate $1e - 2$ for grid encoder, $1e - 3$ for color network.

**Stage 3: 3D mesh refinement** We continue finetuning the 3D mesh from stage 2. Remark that SDXL requires high memory and thus it takes much longer time in optimization due to its size of U-Net. We find that using timestep spacing of 10 works well for SDXL, which enhances efficiency. We use AdamW optimizer with learning rate $1e - 2$ for grid encoder, $1e - 3$ for color network.

## D.2    EVALUATION METRICS

## D.3    HUMAN SURVEY

For each baseline DreamFusion (Poole et al., 2022), Magic3D (Lin et al., 2023), and Prolific-Dreamer (Wang et al., 2023b), we construct 20 binary pairs to rank with our results. We collected 10 binary ratings for each pair, total 600 human ranking were collected. To properly evaluate the quality of 3D content, we ask the following to the human raters:

1. **3D consistency** Which 3D item has a more plausible and consistent geometry?
2. **Prompt fidelity** Which video or images best respects the provided prompt?
3. **Photorealism** Which video or images has a more photorealistic details?

For the 3D consistency, users were asked to select the item that has more geometric consistency, i.e., better shape among different views. For prompt fidelity, users were asked to select the item that is more relevant to the meaning of text prompt. For the photorealism, users were asked to select the item that has more detailed textures that resembles the real 3D objects. The users can select unknown or tie, to refrain their selection.

## D.4    CLIP R-PRECISION.

CLIP R-Precision measures the fidelity of the image in relation to its text input by using CLIP image-text retrieval. For the default CLIP R-precision measure, the distractors, i.e., the captions that distracts the prompts to be retrieved easily, were constructed. We found that the prompts in DreamFusion gallery share similar objects and attributes (e.g., object made out of salad, made out of wood, etc), we use 397 prompts as the full prompt set. During evaluation, we render the albedo images of a 3D scene for 120 uniformly distributed azimuth angles with elevation angle of 15. The camera distances were 1.2 as default. Then we compute CLIP score for the whole images and the full text prompts, and compute the precision with $R = 1$.

