# OpenReview forum: "DreamFlow: High-quality text-to-3D generation by Approximating Probability Flow"
_ICLR.cc/2024/Conference — ICLR 2024 spotlight_

### Official Review · Reviewer_paN1 · 2023-10-28

**Soundness:** 3 good
**Presentation:** 2 fair
**Contribution:** 2 fair
**Rating:** 6
**Confidence:** 4

**Summary:**

The paper presents a novel approach to text-to-3D generation named DreamFlow, which is an optimization-based method similar to score distillation-based methods (e.g., DreamFusion, Magic3D, ProlificDreamer). By reframing text-to-3D optimization as a multi-view image-to-image translation problem, the authors propose a new gradient update rule to drive the optimization. Besides, through the use of a predetermined timestep schedule, the algorithm improves the speed and quality. The paper claims that DreamFlow is 5 times faster and produces higher quality 3D content than existing methods.

**Strengths:**

1. This paper offers a different perspective on how to model 3D generation problems. The authors interpret text-to-3D optimization as a multi-view image-to-image translation problem and propose a solution by approximating the probability flow.
2. The final experimental results of this work are fairly good. In terms of the same quality, it is five times faster than VSD.

**Weaknesses:**

1. Lack of Technical Contribution:
(1). Although the authors are developing an optimization loss from a different angle, the algorithm they ultimately present is actually the same as Variational Score Distillation (VSD). Similarly, I also noticed that the results obtained by the authors are similar to those of VSD. From this perspective, it doesn't seem sufficiently novel.
(2). The author mentions that score distillation uses randomly drawn time steps at each update. However, there have already been many works that use a linear time schedule, which decreases from large t to small t during optimization. Additionally, there is a specialized work (DreamTime) that specifically studies this point. From this perspective, the technique does not appear to be original or novel.

2. Unclear Connection Between Motivation and Technical Details:
(1). From the abstract and introduction, it is evident that the authors' motivation is to address the issue of gradient variance. However, the methods section does not clearly explain how the proposed method or modeling perspective is related to the problem of gradient variance.
They only visualize the gradient norm in the experimental section at the end to indicate that it is more stable (of "norm", instead of "direction"). However, this also relates to the time schedule; a random time schedule would naturally result in varying magnitudes of the gradient because the intensity of diffusion output scores can vary based on the timestep t.
(2). It remains unclear why this ("norm") variance is problematic for the optimization process. How is this demonstrated? For optimization, the direction of the gradient seems to be more important than its magnitude. In the experimental section on page 8, the author states, "This is because of the randomly drawn timestep during optimization, which makes optimization unstable. Thus, we observe that VSD often results in poor geometry during optimization, or over-saturated." I am unclear about the logical relationship indicated by "thus" here.

**Questions:**

1. Why is the variance in the norm of the gradient (which you describe as instability) a detrimental factor? Is there a rigorous and detailed mathematical explanation for this?

2. How is the time schedule related to your modeling? Is your model only applicable to a linear schedule?
What distinguishes your method from the timestep annealing mentioned in works like DreamTime?

3. Does your modeling approach have any broader applications? For instance, are there applications that could not be achieved in previous score distillation methods but might be feasible with your model?

4. Please elaborate on the differences and similarities between your method and VSD, as the final formulations appear to be nearly identical.

5. Please elaborate in detail on where the improvements in experimental results come from. For example, why is it faster, why do you think the results are better than VSD, and have you conducted independent ablation studies? (The current ablation study seems to be of limited significance, as adding each stage naturally leads to better results, but this is not the focus or contribution of the paper).

---

> ### Author Response · Authors · 2023-11-17
> **Response to Reviewer paN1**
>
> Dear reviewer paN1,
>
> We sincerely appreciate your thoughtful comments and suggestions in reviewing our manuscript. We address each of your questions and concerns individually below. Please let us know if you have any comments/concerns that we have not addressed to your satisfaction.
>
> ---
> **[W1, Q2, Q4] Technical Contribution: Difference vs. VSD**
>
> To clarify the difference between APFO and VSD, we provide a detailed explanation in Common Response 1. We also show how APFO differs from VSD with a time-step annealing schedule, either using a linearly decreasing schedule or using weighting priors such as DreamTime [1].
>
> ---
> **[W2, Q1] Motivation and Technical Details**
>
> We provide additional information on how the loss and gradient norms are implied in Common Response 2. At a high level, the convergence of the loss implies the convergence of the distribution to the data distribution, and the low-variance gradient norm implies the stability of the 3D optimization. We also show why high-variance loss and gradient norm in VSD could be problematic in Common Response 2, as this can lead to inconsistent 3D optimization and content shifting during optimization.
>
> ---
> **[Q3] Broader application**
>
> We demonstrate the advantage of our method that it achieves faster 3D optimization when using text-to-image diffusion priors. Our approach could be further applied to various applications in using the diffusion model priors to optimize visual generators (e.g., neural fields), such as text-to-video generation by optimizing video neural fields with video diffusion models, or text-to-4D generation using image and video diffusion models. We believe that exploring APFO for these tasks are interesting directions that we leave for future work.
>
> Furthermore, since our approach makes use of probability flow ODE, it could be more advantageous when using fast-sampling diffusion models such as distilled diffusion models (e.g., guided distillation [2] or consistency models [3]). On the other hand, the current score distillation method cannot achieve full advantage when used with these models.
>
> ---
> **[Q5] Why is the proposed method superior to VSD?**
>
> As we discussed in Common Response 1, our approach allows for faster optimization as we approximate the probability flow ODE, which is a straight path towards data distribution, while VSD utilizes random timesteps in optimization. Also, in Common Response 2, we show that APFO achieves convergence to the data distribution with more stable 3D optimization, while VSD does not. Therefore, those explanations demonstrate how APFO presents faster and higher-quality text-to-3D optimization. Those claims are empirically supported in Tables 2 and 3 of our manuscript, where we provide experiments to compare APFO and VSD under the same setup and show that our approach presents better quality in the CLIP precision metric.
>
> ---
> Reference
>
> [1] Huang, Yukun, et al. "DreamTime: An Improved Optimization Strategy for Text-to-3D Content Creation." arXiv preprint arXiv:2306.12422 (2023).
>
> [2] Meng, Chenlin, et al. "On distillation of guided diffusion models." Proceedings of the IEEE/CVF Conference on Computer Vision and Pattern Recognition. 2023.
>
> [3] Song, Yang, et al. "Consistency models." (2023).

---

> > ### Comment · Reviewer_paN1 · 2023-11-17
> > **Thanks for the response**
> >
> > Thank you for your response. I have increased my score to 5.
> >
> > 1. To ensure the reproducibility of your results, I am curious about the loss weights in your overall loss since score-distillation-based methods are known to be highly sensitive in parameters.
> >
> > 2. Additionally, I would like to inquire whether your loss formulation offers any improvement in addressing the Janus problem. Theoretically, it seems unable to solve this issue, but I am asking out of curiosity, not as a judgment.
> >
> > 3. Regarding the clip score and R-precision, did you generate 100 results and use 397 prompts as reference captions for the calculation?
> >
> > 4. Is it possible to provide a full video gallery of all your generated results used in your evaluation?
> >
> > If these questions are adequately addressed, I am open to considering raising my score to 6 or even 8 points, depending on the responses.

---

> ### Author Response · Authors · 2023-11-18
> **Response to Reviewer paN1**
>
> We are glad to hear that we have addressed your concerns. Regarding your additional questions, we respond to your questions one-by-one in what follows:
>
> ---
> **[Q1] Loss weights**
>
> The hyperparameters are not that sensitive in our framework. We use APFO loss with orientation regularization loss (see Appendix D), i.e., $L = \lambda_{APFO} L_{APFO}+\lambda_{ori}L_{ori}$, where $\lambda_{APFO}$ and $\lambda_{ori}$ are loss weights. We use $\lambda_{APFO}=1.0$ and $\lambda_{ori}=5.0$ throughout all experiments (unless stated otherwise). Here, we found that using $\lambda_{ori}\in [0.0, 10.0]$ works well, while too large orientation loss causes the shape to be over-smoothed. In 3D mesh tuning, we do not use any regularization loss. The details on the optimizers and learning rates are described in Appendix D.
>
> ---
> **[Q2] Janus’ problem**
>
> Since we use pre trained text-to-image diffusion models that do not have 3D knowledge, Janus' problem is still present in our method. One approach in handling Janus’ problem would be using 3D-aware multi-view diffusion models (e.g., concurrent work MVDream [1]), where our proposed method is also applicable to such diffusion models for 3D optimization.
>
> ---
> **[Q3] CLIP R-precision**
>
> Yes, we use 397 prompts as reference captions in computation of CLIP R-precision metric.
>
> ---
> **[Q4] Providing additional results**
>
> Upon your request, we provide 100 3D contents by using different prompts, which we used for evaluation. Please check out on the following link:
>
> **https://drive.google.com/drive/folders/1H1QRb2R0y-gKuAP0LUYd-Px3nL6uAlun?usp=drive_link**
>
> ---
>
> If you have any further questions/concerns, please do not hesitate to let us know !
>
> Thank you very much,
> Authors
>
> Reference
>
> [1] Shi, Yichun, et al. "Mvdream: Multi-view diffusion for 3d generation." arXiv preprint arXiv:2308.16512 (2023).

---

> ### Author Response · Authors · 2023-11-21
> **Reminder**
>
> Dear Reviewer paN1,
>
> Once again, thank you for taking the time to review our paper. We appreciate your efforts in helping us improve our work. As we approach the end of the discussion period in two days, we wanted to inquire if you have any remaining concerns that we could address. Kindly let us know.
>
> Thank you.

---

> > ### Comment · Reviewer_paN1 · 2023-11-21
> >
> > Thanks for the authors' response, I am willing to increase my rating to 6.

---

> > > ### Author Response · Authors · 2023-11-22
> > > **Response**
> > >
> > > We appreciate that our rebuttal addressed your concerns. Also, thank you for raising the score!
> > >
> > > Please let us know if you have any further questions.

---

### Official Review · Reviewer_ZxmY · 2023-10-29

**Soundness:** 2 fair
**Presentation:** 3 good
**Contribution:** 3 good
**Rating:** 8
**Confidence:** 3

**Summary:**

This paper proposes a new method for text-to-3D from pre-trained 2D diffusion model. It formulates the 3D generation problem as a Schrodinger Bridge problem that transports multi-view renderings to the data distribution defined by the 2D diffusion model. An amortized sampling scheme is also proposed to avoid using random time step sampling during optimization. The method has more stable optimization loss variation and yields better quality results than previous methods, according to user preference study.

**Strengths:**

1. The formulation of text-to-3D as solving a Schrodinger Bridge problem is a novel perspective and insightful.

2. The proposed amortized sampling method stabilizes training loss (Figure 6).

3. User study is conducted to showcase the effectiveness of the method, which is more convincing than other quantitative metrics.

**Weaknesses:**

Although motivated differently, the final ODE (equation 12) looks very similar to VSD (equation 8). It looks like the main benefit of more stable training (Figure 6) comes from the amortized sampling (Section 3.2) where there is an inner loop for each t and t decreases gradually. I think the same thing can be implemented for VSD as well, but no such comparison is shown. Maybe this sampling method is more naturally motivated for this method than VSD. But if the above understanding is correct, I think it would be better if the authors can be more explicit about the similarity and difference between the two algorithms.

**Questions:**

No questions.

---

> ### Author Response · Authors · 2023-11-17
> **Response to Reviewer ZxmY**
>
> Dear Reviewer ZxmY,
>
> We sincerely appreciate your thoughtful comments and suggestions in reviewing our manuscript. We address each of your questions and concerns individually below. Please let us know if you have any comments/concerns that we have not addressed to your satisfaction.
>
> ---
> **[W1] Comparison to VSD and VSD to other timestep annealing methods**
>
> In Common Response 1, we detail the difference between VSD and APFO. Also, we compare APFO and VSD with timestep annealing methods including DreamTime [1]. As you understood, we remark that APFO is a more naturally motivated approach in using a decreasing timestep schedule, and presents faster optimization with higher fidelity.
>
> ---
> Reference
>
> [1] Huang, Yukun, et al. "DreamTime: An Improved Optimization Strategy for Text-to-3D Content Creation." arXiv preprint arXiv:2306.12422 (2023).

---

> ### Author Response · Authors · 2023-11-21
> **Reminder**
>
> Dear Reviewer ZxmY,
>
> Once again, thank you for taking the time to review our paper. We appreciate your efforts in helping us improve our work. As we approach the end of the discussion period in two days, we wanted to inquire if you have any remaining concerns that we could address. Kindly let us know.
>
> Thank you.

---

> > ### Comment · Reviewer_ZxmY · 2023-11-22
> >
> > Thanks for the rebuttal. Increased my rating to 8.

---

> > > ### Author Response · Authors · 2023-11-22
> > > **Response**
> > >
> > > We appreciate that our rebuttal addressed your concerns. Also, thank you for raising the score!
> > >
> > > Please let us know if you have any further questions.

---

### Official Review · Reviewer_pJwX · 2023-10-29

**Soundness:** 3 good
**Presentation:** 3 good
**Contribution:** 2 fair
**Rating:** 6
**Confidence:** 4

**Summary:**

This work designs DreamFlow, a practical three-stage coarseto-fine text-to-3D optimization framework that enables fast generation of highquality and high-resolution (i.e., 1024×1024) 3D contents. The experiments show that the framework outperforms exisiting baelines in both user-study and quantitative results.

**Strengths:**

1. The paper is well-written and the experiments are sufficient.
2. The theoretical part seems sound to me.
3. Some results have good visual quality.

**Weaknesses:**

1. My main concern is that the proposed APFO is too similar with VSD. Maybe a more detailed explanation should be given to demonstrate the difference between VSD and APFO.
2. The color of some results are kind of "green" or "brown" (including the cactus in Fig. 4, the corgi in Fig. 7, etc.), can you explain the reason?

**Questions:**

1. What is the CFG scale used in 1/2/3 stages?
2. How is the GPU memory consumation of finetuning the LoRA of SDXL in stage3?
3. Can you provide the 2D experiment of APFO (using APFO to optimize a 2D image)? A 2D experiment will demonstrate the effectiveness of APFO better.

---

> ### Author Response · Authors · 2023-11-17
> **Response to Reviewer pJwX**
>
> Dear reviewer pJwX,
>
> We sincerely appreciate your thoughtful comments and suggestions in reviewing our manuscript. We address each of your questions and concerns individually below. Please let us know if you have any comments/concerns that we have not addressed to your satisfaction.
>
> ---
> **[W1] Comparison to VSD**
>
> We clarify the difference between APFO and VSD in Common Response 1. We remark that the APFO leverages the time-derivative of noise scale derived from probability flow ODE in leveraging the diffusion model prior, as opposed to the weighting function in VSD. This leads to faster convergence and better fidelity as shown in 2D experiments.
>
> ---
> **[W2] Rationale for Color Bias**
>
> Note that the color bias does not exist in our method, and different colors or textures can be generated due to the diversity of generation. However, as we have shown in the limitation part in Section 6, some undesirable results could be obtained due to the use of pre-trained text-to-image diffusion models.
>
> ---
> **[Q1] CFG Scale**
>
> For your information, we use a classifier-free guidance scale of 7.5 for each stage.
>
> ---
> **[Q2] Memory consumption**
>
> For your information, our approach requires >30GB for stage 1 NeRF optimization, and less than 20GB for stages 2 and 3 3D mesh tuning. Note that since we use instant NGP for stage 1 NeRF optimization, the GPU memory may vary due to sparsity during optimization. We tested with an A100 40GB GPU.
>
> ---
> **[Q3] 2D experiments**
>
> Following your suggestion, we provide 2D experiments to compare APFO with VSD and VSD with timestep annealing methods (see Common Response 1 and Appendix B of revised manuscripts for details). Note that APFO can produce high-fidelity 2D images with fewer steps than VSD without blurry artifacts.

---

> ### Author Response · Authors · 2023-11-21
> **Reminder**
>
> Dear Reviewer pJwX,
>
> Once again, thank you for taking the time to review our paper. We appreciate your efforts in helping us improve our work. As we approach the end of the discussion period in two days, we wanted to inquire if you have any remaining concerns that we could address. Kindly let us know.
>
> Thank you.

---

> > ### Comment · Reviewer_pJwX · 2023-11-21
> > **Thank you for your rebuttal.**
> >
> > The rebuttal partially solves my question. I think it's interesting to train the LoRA of SDXL. I suggest you include more implementation details about training with SDXL and how it's compared to SD.

---

> > > ### Author Response · Authors · 2023-11-22
> > > **Response**
> > >
> > > We appreciate that we have addressed most of your concerns. Regarding your remaining question, the training for SDXL and SD LoRA is the same, except for different hyperparameters such as learning rate and LoRA rank. For further information, please refer to Appendix D.1.

---

### Official Review · Reviewer_9SLV · 2023-10-31

**Soundness:** 2 fair
**Presentation:** 3 good
**Contribution:** 2 fair
**Rating:** 5
**Confidence:** 4

**Summary:**

The authors propose a three-stage coarse-to-fine text-to-3D generation pipeline capable of producing high-resolution 3D assets from given text prompts. This involves the use of a variant of the SDS loss, Approximate Probability Flow ODE (APFO), to optimize a 3D representation learning from text-to-image priors. NeRFs and Meshes are used as the 3D representations at different stages, with the SDXL refiner applied to further enhance the extracted 3D meshes.

**Strengths:**

The paper is well-written and easy to follow. The perspective of considering SDS as a Schrödinger Bridges (SB) problem is insightful. The visual results appear comparable, and the quantitative scores are slightly better to the baselines. Notably, the proposed method demonstrates faster convergence than state-of-the-art methods.

**Weaknesses:**

1) While the authors provide insightful interpretations of the SDS method as an SB problem, its implementation is quite similar to the prior work ProlificDreamer. Additionally, timestep scheduling has also been proposed in previous works [1, 2, 3], limiting the impact of the proposed method.

2) When viewed as an SB problem, the proposed method using a LoRA-fine-tuned score function seems to have difficulties in generating diverse results. Further explanations are encouraged.

3) The improved numerical scores, as shown in the paper, are modest and are computed based on only 20 3D assets. Furthermore, some results contain weird textures, as observed in the example of "A stack of pancakes covered in maple syrup."

4) The results after the third stage exhibit color bias (Fig.5), and most generated 3D assets do not appear significantly improved after SDXL fine-tuning, except for the cottage example in Fig. 13. To explain this, additional ablation studies are recommended to fully evaluate the effectiveness of SDXL.

Miscellaneous:

1) An explanation of the method referred to as "tie" is missing in Table 1.

2) There appear to be typos in Eq.13, with "\Phi" possibly intended to be "\phi," and "q" possibly intended to be "p." Otherwise, clarification regarding the transition from Eq.12 to Eq.13 would be appreciated.


[1] ProlificDreamer: High-Fidelity and Diverse Text-to-3D Generation with Variational Score Distillation, Wang et al., NeurIPS 2023.

[2] HiFA: High-fidelity Text-to-3D with Advanced Diffusion Guidance, Zhu et al., 2023.

[3] DreamTime: An Improved Optimization Strategy for Text-to-3D Content Creation, Huang et al., 2023.

**Questions:**

1) In Fig.6, the stability of the APFO loss is attributed to the non-increasing timestep scheduling, as opposed to the two-stage stratified random scheduling used in the VSD loss from ProlificDreamer. However, the convergence of the loss curves may not precisely reflect the training quality of the 3D representation due to the random timestep sampling in VSD. Could the authors provide further clarification on this point?

2) Based on the above question, what would be the outcome if the VSD loss were applied using the same timestep scheduling as proposed by the authors? Moreover, the predetermined timestep selection process remains unclear, particularly regarding the decreasing ratio over iterations.

**Details Of Ethics Concerns:**

The generation of realistic and high-fidelity 3D assets may potentially raise intellectual property (IP) concerns.

---

> ### Author Response · Authors · 2023-11-17
> **Response to Reviewer 9SLV**
>
> Dear Reviewer 9SLV,
>
> We sincerely appreciate your thoughtful comments and suggestions in reviewing our manuscript. We address each of your questions and concerns individually below. Please let us know if you have any comments/concerns that we have not addressed to your satisfaction.
>
> ---
> **[W1] Difference between VSD and comparison with previous works on decreasing noise schedule**
>
> In Common Response 1, we clarify that APFO and VSD have different optimization objectives, and they are different in implementation as APFO is entailed with a predetermined decreasing timestep schedule. In addition, APFO not only differs from VSD together with a decreasing timestep schedule, but also is a better approach in leveraging the diffusion model prior. We also compare APFO with VSD + DreamTime [1], which uses priors on the weighting function. Our explanation with 2D experiments shows that APFO is not only different from VSD with timestep annealing, but also faster (in terms of optimization speed) and better at generating high-fidelity images.
>
> ---
> **[W2] Regarding the Diversity**
>
> Our method does not suffer from diversity since we use small CFG scales (e.g. $\omega=7.5$) throughout the experiment.
>
> ---
> **[W3] Experimental Validation**
>
> We conducted a human evaluation with 20 text prompts to compare with other baselines, where 10 pairwise comparisons were made for each prompt, thus 200 comparisons were made for each baseline. Our human evaluation experiment suggests that our method shows better photorealism than previous work. In fact, in Table 2 and Table 3, we perform an additional comparison with 100 text prompts taken from the DreamFusion gallery, which shows that our method outperforms others in the CLIP R-precision metric.
>
> ---
> **[W4] Additional ablation study on the effect of using SDXL**
>
> To further demonstrate the effectiveness of Stage 3 mesh refinement, we provide more qualitative results in Appendix Figure 13 of our revised manuscript. Remark that Stage 3 mesh refinement results in more high-contrast 3D contents. Also, for quantitative evaluation, we conduct an ablation study by computing CLIP aesthetic score [2]. Similar to evaluation of CLIP R-precision, we render views and compute CLIP aesthetic scores and report average scores. Note that the average CLIP aesthetic score has improved after stage 3 mesh refinement from 4.82 to 5.02, which indicates the advantage of stage 3 mesh refinement.
> Nevertheless, as we have shown in the limitation part in Section 6, some undesirable results could be obtained due to the use of pre-trained text-to-image diffusion models.
>
>
> ---
> **[Q1] Gradient variance and the quality of 3D representation**
>
> To clarify our points, in Common Response 2, we provide detailed analysis on how the convergence of loss and gradient norm indicates in 3D optimization. Indeed, the convergence of loss denotes the convergence of 3D views to the data distribution and the low-variance gradient norm indicates the stability of 3D optimization, where both implies superiority of APFO compared to VSD.
>
> ---
> **[Q2-1] Comparison with VSD + timestep annealing**
>
> Please refer to [W1] and Common Response 1 for detailed comparison on APFO with VSD and its variants with timestep annealing methods including DreamTime [1].
>
> **[Q2-2] Timestep selection**
> To clarify, we choose the time steps $t_1,\cdots,t_N$ such that they have decreasing variance $\sigma(t_1) >\cdots > \sigma(t_N)$. In general, diffusion models are trained with 1000 timesteps, and we denote the timestep as the ratio divided by 1000 (e.g., $t_1=1.0$ denotes the timestep at 1000, and $t_N=0.2$ denotes the timestep at 200). While we use $t_1=1.0$ when optimizing NeRF from scratch (i.e., step 1), we choose a smaller value of $t_1$ for fine-tuning the 3D mesh (i.e., steps 2 and 3). This is because starting from a large time step may alter the textures due to the addition of large amounts of noise and lose fidelity to the original 3D content from the previous stage. Empirically, we have found that $t_1=0.5$ is sufficient for stage 2 mesh refinement and $t_1=0.3$ is sufficient for stage 3 mesh refinement.
>
> ---
> **[Miscellaneous]**
>
> Thank you for pointing out the typos and missing details. We will address these before the discussion period closes. Regarding the details in Eq. (12) to Eq. (13), it simply computes a gradient $dx/dt$ by inserting discrete time steps, and the loss is given by the difference between the generator $g(\theta,c)$ and the target generated by $x+\Delta t \cdot dx/dt$.
>
> ---
> Reference
>
> [1] Huang, Yukun, et al. "DreamTime: An Improved Optimization Strategy for Text-to-3D Content Creation." arXiv preprint arXiv:2306.12422 (2023).
>
> [2] C Schuhmann. Laoin aesthetic predictor. 2022. URL https://laion.ai/blog/ laion-aesthetics/.

---

> ### Author Response · Authors · 2023-11-21
> **Reminder**
>
> Dear Reviewer 9SLV,
>
> Once again, thank you for taking the time to review our paper. We appreciate your efforts in helping us improve our work. As we approach the end of the discussion period in two days, we wanted to inquire if you have any remaining concerns that we could address. Kindly let us know.
>
> Thank you.

---

### Author Response · Authors · 2023-11-17
**Common Response**

Dear reviewers and AC,

We sincerely appreciate your efforts spent reviewing our manuscript.

As reviewers highlighted, we believe our work demonstrates a novel idea in text-to-3D generation (9SLV, ZxmY, paN1) with theoretical principle (pJwX, ZxmY, paN1) validated with good qualitative examples (pJwX, paN1).
We appreciate reviewers’ constructive comments on our manuscript. In response, we carefully revised and enhanced the manuscript with following:
- Details in comparison with prior works, e.g., VSD (Appendix B)
- 2D experiments to demonstrate the effectiveness of APFO (Figure 14)
- More qualitative results showing the effectiveness of stage 3 mesh refinement (Figure 13)

These updates are temporarily highlighted in “red” for your convenience.

Furthermore,  we collected the common questions that multiple reviewers have asked and responded to each question one-by-one in what follows.

---
**Common response 1. Comparison between VSD and APFO**

First, we emphasize that the optimization objectives of APFO (ours) and VSD (prior work) are different even when used with the same timstep schedule of APFO. In specific, as in Eq. (12), APFO scales the difference of denoiser outputs by time-derivative of noise variance, i.e. $\frac{\dot \sigma(t)}{\sigma(t)}$ (with the predetermined timesteps), which exactly amounts the interval to transport latents to the next noise level. In contrast, VSD uses the (empirically chosen) weighting function $\lambda(t)=\sigma^2(t)$ (with randomly chosen timesteps). The difference in training objectives, $\frac{\dot \sigma(t)}{\sigma(t)}$ vs. $\lambda(t)=\sigma^2(t)$, is crucial; even when VSD is employed with a decreasing noise schedule as like APFO, it cannot achieve the main goal of APFO to approximate probability flow ODE, i.e., to emulate the generative process of the diffusion model so that the optimized image follows the data distribution.

For better explanation, we provide a 2D image generation experiment that compares APFO, standard VSD (i.e., random timestep sampling), and VSD with timestep annealing (linearly decreasing as like APFO). Furthermore, as requested by several reviewers, we show an additional comparison with DreamTime [1], which proposes to sample timesteps according to predetermined weighting functions. The results are shown in Appendix B Figure 14 of the revised manuscript. One can observe that the image generated by APFO shows higher fidelity, while other methods show blurry artifacts. Also, the blurry artifact is not reduced even when using an annealing timestep schedule for VSD or VSD scheduled with DreamTime. Also note that APFO produces a more faithful image even when using 200 optimization steps, while other methods use 500 steps. This is because APFO approximates probability flow ODE, which is a straight path towards data distribution, while VSD traverses over random timesteps that prolongs the optimization. This 2D experiment shows that 1) APFO is not only different from simply taking VSD with timestep annealing, but also presents a more faithful image, and 2) by approximating the probability flow of a pre-trained diffusion model, APFO is able to generate an image with significantly fewer optimization steps.

---
**Common response 2. Regarding loss and gradient norm analysis**

In Figure 6 of our manuscript, we plot losses and gradient norms during optimization to show how APFO enables better 3D optimization than VSD. To convey better implication of this figure, first note that the APFO loss in Eq. (13) is equal to the norm of the time-derivative $dx / dt$, i.e.
$$
L = || \frac{\dot\sigma(t)}{\sigma(t)} (D_\phi (x;\sigma(t)) - D_q(x;\sigma(t)||_2^2 \approx || \nabla_x \log p(x;\sigma(t)) - \nabla_x \log q(x;\sigma(t))||_2^2,
$$
where the latter term is equivalent to the Fisher divergence between data distribution and distribution of images from generators. Thus, the convergence of loss indicates the convergence of distribution with respect to Fisher divergence. Also, note that the gradient norm is given by
$$
|| \partial L / \partial \theta || = L ||\partial {\hat{x}} / \partial \theta||,
$$
where $\partial{\hat{x}} / \partial \theta$ is the gradient norm of the generator. Therefore, the convergence of gradient norm indicates the stability of 3D optimization. On the other hand, VSD shows high variance of the loss and gradient norm during the optimization, which is due to the randomly chosen timesteps. We remark that this can be problematic because, first, the high variance implies inconsistent updating for different views, resulting in inconsistent 3D generation. Second, the high variance of the loss implies that the views are kept updated without convergence, so the contents are shifted during the optimization (e.g., Figure 14).

Reference

[1] Huang, Yukun, et al. "DreamTime: An Improved Optimization Strategy for Text-to-3D Content Creation." arXiv preprint arXiv:2306.12422 (2023).

---

### Meta-Review · Area_Chair_WWJM · 2023-12-06

**Metareview:**

The paper proposes a framework fro coarse-to-fine text-to-3D generation with 2D diffusion model as a prior. The paper received one negative and three positive ratings. The positive reviews were based on the novel and effective formulation. Also, both theoretical soundness and qualitative results are provided. The negative reviews concerned the comparisons to existing methods (VSD, specifically) and additional visual results of the mesh refinement stage, which in my opinion were well addressed during the rebuttal. We thus recommend the acceptance of the paper.

**Justification For Why Not Higher Score:**

Despite the proposed formulation is effective and technically sound, the visual quality is similar to the SoTA results.

**Justification For Why Not Lower Score:**

The proposed formulation is novel and well validated.

---

### Decision · Program_Chairs · 2024-01-16

Accept (spotlight)